# ABSORBING QUANTIZATION ERROR BY DEFORMABLE NOISE SCHEDULER FOR DIFFUSION MODELS

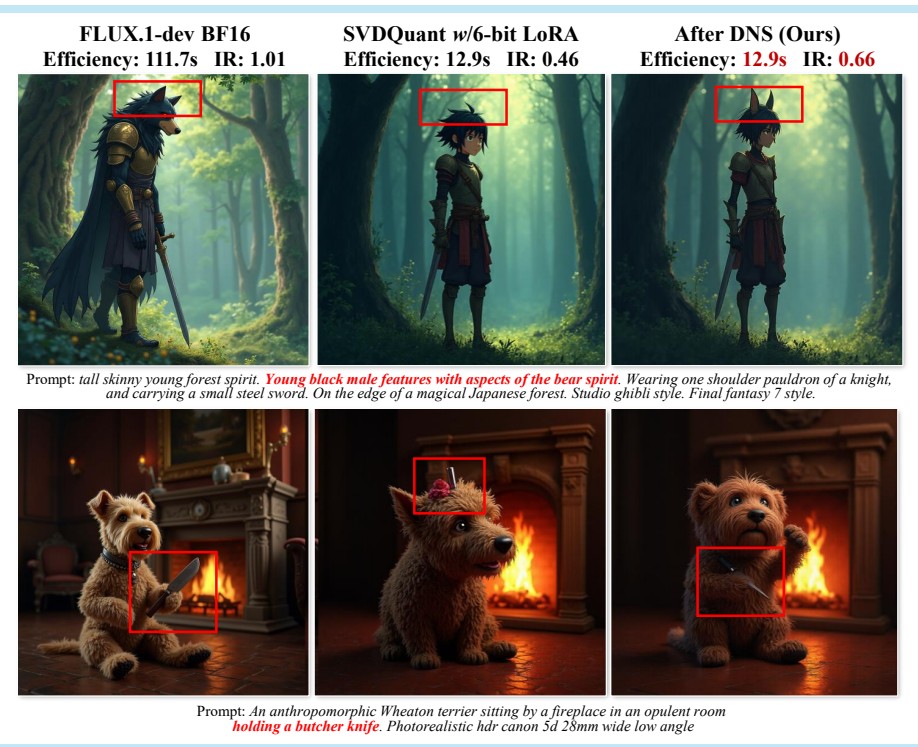

Prompt: *tall skinny young forest spirit.* ***Young black male features with aspects of the bear spirit****. Wearing one shoulder pauldron of a knight, and carrying a small steel sword. On the edge of a magical Japanese forest. Studio ghibli style. Final fantasy 7 style.*

Prompt: *An anthropomorphic Wheaton terrier sitting by a fireplace in an opulent room* ***holding a butcher knife****. Photorealistic hdr canon 5d 28mm wide low angle*

Figure 1: Our method offers a plug-and-play approach to improve the quantization performance of diffusion models (including Flow-matching), enabling seamless integration with existing PTQ methods without compromising inference speed.

## ABSTRACT

Diffusion models deliver state-of-the-art image quality but are expensive to deploy. Post-training quantization (PTQ) can shrink models and speed up inference, yet residual quantization errors distort the diffusion distribution (the timestep-wise marginal over $x_t$), degrading sample quality. We propose a distribution-preserving framework that absorbs quantization error into the generative process without changing architecture or adding steps. (1) Distribution-Calibrated Noise Compensation (DCNC) corrects the non-Gaussian kurtosis of quantization noise via a calibrated uniform component, yielding a closer Gaussian approximation for robust denoising. (2) Deformable Noise Scheduler (DNS) reinterprets quantization as a principled timestep shift, mapping the quantized prediction distribution $x_t$ back onto the original diffusion distribution so that the target marginal is preserved. Unlike trajectory-preserving or noise-injection methods limited to stochastic samplers, our approach preserves the distribution under both stochastic and deterministic samplers and extends to flow-matching with Gaussian conditional paths. It is plug-and-play and complements existing PTQ schemes. On DiT-XL (W4A8),

our method reduces FID from 9.83 to 8.51, surpassing the FP16 baseline (9.81), demonstrating substantial quality gains without sacrificing the efficiency benefits of quantization.

# 1 INTRODUCTION

Diffusion models have become the leading paradigm for high-quality generative modeling, powering applications across image synthesis (Ho et al., 2020; Rombach et al., 2022; Labs, 2024), style transfer (Qi et al., 2024), video generation (Brooks et al., 2024), 3D reconstruction (Zhou et al., 2024), and robotics (Chi et al., 2024). Their rapid scaling– from Stable Diffusion's 800M parameters (Rombach et al., 2022) to SDXL's 2.6B (Podell et al., 2023), and most recently to FLUX.1's 12B (Labs, 2024)– has driven remarkable improvements in fidelity and controllability. However, these gains come at a steep computational cost: inference with large models demands tens of gigabytes of GPU memory and seconds per image, limiting deployment in real-time and resource-constrained settings.

Model quantization is a practical solution to this bottleneck. By reducing weights and activations from floating-point to low-bit integer, quantization compresses model size and accelerates inference (Jacob et al., 2017; Nagel et al., 2021). Recent post-training quantization (PTQ) methods have successfully reduced diffusion models to 8, 4 or even 1-bit precision (Li et al., 2023; Wu et al., 2024; Li et al., 2025); (Huang et al., 2024; Shang et al., 2023; Ye et al., 2024; Wang et al., 2024; Zheng et al., 2024). Yet, unlike discriminative networks or autoregressive language models, diffusion models are especially sensitive to quantization. Errors accumulate across iterative denoising steps, corrupting the assumed Gaussian noise distribution and significantly degrading sample quality (He et al., 2023).

Existing approaches attempt to mitigate this challenge by either reshaping weight distributions to reduce quantization error (Li et al., 2023; Wu et al., 2024; Li et al., 2025) or injecting approximated Gaussian noise to absorb residual errors (He et al., 2023). However, these methods have important limitations: weight-reshaping strategies only preserve stepwise predictions without addressing distributional drift, while noise-injection methods are restricted to stochastic samplers and fail on deterministic or flow-matching frameworks (Lipman et al., 2023). Moreover, quantization noise rarely follows an ideal Gaussian distribution, undermining the robustness of Gaussian-based corrections.

In this work, we introduce a distribution-preserving quantization framework that directly absorbs quantization error into the diffusion process. Our approach consists of two key components: (i) *Distribution-Calibrated Noise Compensation (DCNC)*: corrects the non-Gaussian kurtosis of quantization noise through a calibrated uniform component, yielding a more faithful Gaussian approximation. (ii) *Deformable Noise Scheduler (DNS)*: reinterprets quantization effects as a timestep shift in the diffusion process, aligning quantized prediction with the original diffusion distribution without modifying model architecture or adding inference overhead. This framework preserves the diffusion distribution under quantization, making it compatible with both stochastic and deterministic samplers as well as flow-matching models. It operates as a plug-and-play solution,

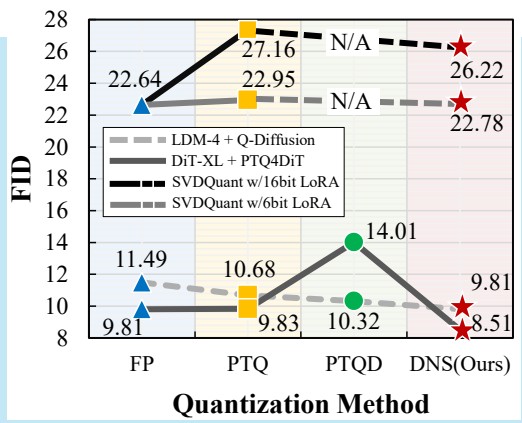

Figure 2: Comparison of FID scores across quantization methods for different backbones.

easily integrated with existing PTQ methods. Empirically, our method consistently improves generation quality across diverse backbones (see Fig. 2). For example, on DiT-XL our method reduces the FID of PTQ4DiT (W4A8) from 9.83 to 8.51, surpassing even the FP16 baseline (9.81).

In summary, our contributions are threefold: (1) A principled analysis showing that quantization can be reinterpreted as timestep shifts in the diffusion process. (2) A deformable noise scheduler and distribution-calibrated correction that preserve the generative distribution under quantization. (3)

Extensive experiments across LDM, DiT, and FLUX demonstrating consistent quality improvements without sacrificing the efficiency benefits of quantization.

## 2 RELATED WORK

**Diffusion Models.** Diffusion models (Sohl-Dickstein et al., 2015; Ho et al., 2020) were originally formulated as stochastic processes for data generation. Denoising Diffusion Implicit Models (DDIM) (Song et al., 2022) later reduced the number of sampling steps, significantly accelerating the generation process. More recently, Flow Matching (Lipman et al., 2023) has been introduced as a deterministic alternative, powering state-of-the-art systems such as FLUX (Labs, 2024).

The backbone architecture of early diffusion models was primarily based on UNets (Rombach et al., 2022). Diffusion Transformers (DiT) (Peebles & Xie, 2023) pioneered the use of pure transformer architectures, leading to improved scalability and performance. Large-scale text-to-image models such as Stable Diffusion (Rombach et al., 2022), SDXL (Podell et al., 2023), PixArt (Chen et al., 2023), and FLUX (Labs, 2024) have since revolutionized content generation by pushing model sizes larger to achieve better fidelity, diversity, and controllability.

**Model Quantization for Diffusion models.** Quantization reduces computational costs by representing model parameters and activations with lower-precision formats. Approaches generally fall into two categories: quantization-aware training (QAT) (Jacob et al., 2017), which requires retraining with quantization in the loop, and post-training quantization (PTQ) (Nagel et al., 2021), which adapts pre-trained models with minimal calibration data. Due to the size and training cost of diffusion models, PTQ methods are more practical and thus widely adopted.

Specialized PTQ techniques have been developed to address the unique challenges of diffusion backbones. Q-Diffusion (Li et al., 2023) employs channel-wise quantization with explicit outlier handling. PTQ4DM (Shang et al., 2023) and PTQ4DiT (Wu et al., 2024) design quantization strategies tailored to their respective UNet and transformer backbones. MixDQ (Zhao et al., 2024) adopts mixed precision to balance efficiency and quality. SVDQuant (Li et al., 2025) introduces low-rank branches to capture residual information lost during quantization.

A different line of work, PTQD (He et al., 2023), proposes an "absorbing quantization error" strategy that integrates quantization noise into the random noise term of stochastic samplers. While effective in certain settings, this approach is fundamentally restricted to stochastic sampling and cannot be applied to deterministic samplers or flow-matching frameworks that dominate modern implementations. Moreover, the injected quantization noise often deviates from the expected Gaussian distribution, which can misalign the diffusion process and degrade generation quality.

## 3 PRELIMINARY

### 3.1 DIFFUSION MODELS AND FLOW MATCHING

**Diffusion Models** (Ho et al., 2020; Song et al., 2022) are generative models based on a two-stage process: a fixed *forward* noising process and a learnable *reverse* denoising process.

In the forward process, a clean data sample $\boldsymbol{x}_0 \sim q(\boldsymbol{x}_0)$ is progressively corrupted by Gaussian noise over $T$ timesteps via a Markov chain with a predefined variance schedule $\{\beta_t\}_{t=1}^T$, where $0 < \beta_t < 1$. Define $\alpha_t := 1 - \beta_t$ and $\bar{\alpha}_t := \prod_{i=1}^t \alpha_i$. The forward transition distributions are given by:

$$q(\boldsymbol{x}_t \mid \boldsymbol{x}_{t-1}) = \mathcal{N}\big(\sqrt{\alpha_t}\,\boldsymbol{x}_{t-1},\, \beta_t \mathbf{I}\big), \qquad q(\boldsymbol{x}_t \mid \boldsymbol{x}_0) = \mathcal{N}\big(\sqrt{\bar{\alpha}_t}\,\boldsymbol{x}_0,\, (1 - \bar{\alpha}_t)\mathbf{I}\big). \tag{1}$$

The reverse process aims to recover $\boldsymbol{x}_0$ from $\boldsymbol{x}_T$ by learning a parametric approximation $p_\theta(\boldsymbol{x}_{t-1} \mid \boldsymbol{x}_t)$ to the true reverse distribution. Ho et al. (2020) (DDPM) model this reverse process as a Gaussian with fixed variance, trained to predict the noise added at each step. Song et al. (2022) (DDIM) generalize this by introducing a family of non-Markovian reverse processes parameterized by a noise schedule $\{\sigma_t\}_{t=1}^T$, enabling faster sampling with deterministic or stochastic trajectories.

Specifically, for any $t > 1$, the conditional distribution of $\boldsymbol{x}_{t-1}$ given $\boldsymbol{x}_t$ and clean sample $\boldsymbol{x}_0$ is:

$$q_\sigma(\boldsymbol{x}_{t-1} \mid \boldsymbol{x}_t, \boldsymbol{x}_0) = \mathcal{N}\big(\hat{\mu}_t(\boldsymbol{x}_t, \boldsymbol{x}_0),\, \sigma_t^2 \mathbf{I}\big), \tag{2}$$

where $\sigma_t \geq 0$ controls the stochasticity of the reverse step and $\hat{\mu}_t(\boldsymbol{x}_t, \boldsymbol{x}_0) = \sqrt{\bar{\alpha}_{t-1}}\,\boldsymbol{x}_0 + \sqrt{1 - \bar{\alpha}_{t-1} - \sigma_t^2} \cdot \frac{\boldsymbol{x}_t - \sqrt{\bar{\alpha}_t}\,\boldsymbol{x}_0}{\sqrt{1-\bar{\alpha}_t}}$. When $\sigma_t = \sqrt{(1-\bar{\alpha}_{t-1})/(1-\bar{\alpha}_t)} \cdot \sqrt{1 - \bar{\alpha}_t/\bar{\alpha}_{t-1}}$, this recovers the DDPM posterior; when $\sigma_t = 0$, the process becomes deterministic (DDIM).

Since $\boldsymbol{x}_0$ is unknown during sampling, it is estimated from $\boldsymbol{x}_t$ using a noise-prediction network $\epsilon_\theta(\boldsymbol{x}_t, t)$. The denoised estimate $\hat{\boldsymbol{x}}_0$ is obtained via:

$$\hat{\boldsymbol{x}}_0 = \frac{1}{\sqrt{\bar{\alpha}_t}}\Big(\boldsymbol{x}_t - \sqrt{1 - \bar{\alpha}_t}\,\varepsilon_\theta(\boldsymbol{x}_t, t)\Big). \tag{3}$$

The learned reverse process is then defined by substituting $\hat{\boldsymbol{x}}_0$ for $\boldsymbol{x}_0$ in Eq. 2:

$$p_\theta(\boldsymbol{x}_{t-1} \mid \boldsymbol{x}_t) = q_\sigma(\boldsymbol{x}_{t-1} \mid \boldsymbol{x}_t, \hat{\boldsymbol{x}}_0), \quad t > 1. \tag{4}$$

This leads to the unified sampling update rule:

$$\boldsymbol{x}_{t-1} = \sqrt{\bar{\alpha}_{t-1}}\,\hat{\boldsymbol{x}}_0 + \sqrt{1 - \bar{\alpha}_{t-1} - \sigma_t^2}\,\frac{\boldsymbol{x}_t - \sqrt{\bar{\alpha}_t}\,\hat{\boldsymbol{x}}_0}{\sqrt{1-\bar{\alpha}_t}} + \sigma_t z, \quad z \sim \mathcal{N}(0, \mathbf{I}). \tag{5}$$

**Flow matching** (Lipman et al., 2023) generalizes diffusion via a continuous-time stochastic interpolant between Gaussian noise $\epsilon \sim \mathcal{N}(0, I)$ and data $x_0$. A simple linear interpolant is

$$\boldsymbol{x}_t = (1 - t)\boldsymbol{x}_0 + t\epsilon, \quad t \in [0, 1]. \tag{6}$$

The model learns a velocity field $\boldsymbol{v}_\theta(\boldsymbol{x}_t, t)$ such that the probability flow ODE (PF-ODE)

$$\frac{d\boldsymbol{x}_t}{dt} = \boldsymbol{v}_\theta(\boldsymbol{x}_t, t) \tag{7}$$

induces marginal distributions that match that of Eq. 6 for all time $t \in [0, 1]$. Sampling from the model thus reduces to solving the PF-ODE in Eq. 7 using standard ODE solvers, e.g., Euler integration, starting from Gaussian noise $\varepsilon \sim \mathcal{N}(0, \boldsymbol{I})$ (Lipman et al., 2023), which matches the ground-truth dynamics. Sampling then amounts to integrating this ODE deterministically from noise to data. Flow matching, owing to its high scalability, has already been adopted in modern text-to-image models such as FLUX (Labs, 2024). Flow matching thus unifies diffusion-style objectives with a deterministic transport formulation, and underlies modern large-scale systems such as FLUX (Labs, 2024).

## 3.2 Quantization Noise Correction and Absorption

Quantizing diffusion models inevitably introduces quantization error. This error can be systematically decomposed into two parts: (i) a correlated component that scales linearly with the original network output $\epsilon_\theta(\boldsymbol{x}_t, t)$, and (ii) an independent component $\Delta'\epsilon_\theta(\boldsymbol{x}_t, t)$ that is uncorrelated with the prediction. This decomposition, referred to as Correlation Noise Correction (CNC), can be written as:

$$\Delta\epsilon_\theta(\boldsymbol{x}_t, t) = k\epsilon_\theta(\boldsymbol{x}_t, t) + \Delta'\epsilon_\theta(\boldsymbol{x}_t, t) \tag{8}$$

where $\Delta\epsilon_\theta(\boldsymbol{x}_t, t)$ denotes total quantization error, $k$ is a scalar correlation coefficient, and $\Delta'\epsilon_\theta(\boldsymbol{x}_t, t)$ represents the residual independent component. PTQD (He et al., 2023) approximates $\Delta'\epsilon_\theta(\boldsymbol{x}_t, t)$ as Gaussian noise, but in practice this assumption is often inaccurate. The variance of the independent noise, denoted $\sigma_q^2$, can instead be empirically estimated from representative data samples.

In stochastic sampling process, sampler involves adding random noise in Eq. 5. PTQD absorbs the disentangled independent quantization error into this random term, effectively replacing:

$$\sigma_t z \rightarrow \sigma_t z - \frac{\beta_t}{\sqrt{\alpha_t}\sqrt{1 - \bar{\alpha}_t}(1 + k)}\Delta'\epsilon_\theta(\boldsymbol{x}_t, t) \tag{9}$$

This leads to a modified update rule:

$$\boldsymbol{x}_{t-1} = \frac{1}{\sqrt{\alpha_t}}\left(\boldsymbol{x}_t - \frac{\beta_t}{\sqrt{1-\bar{\alpha}_t}}\epsilon_\theta(\boldsymbol{x}_t, t)\right) + \sigma_t z - \frac{\beta_t}{\sqrt{\alpha_t}\sqrt{1-\bar{\alpha}_t}(1+k)}\Delta'\epsilon_\theta(\boldsymbol{x}_t, t) \tag{10}$$

In summary, PTQD attempts to (i) preserve the injected noise distribution by treating the disentangled quantization residual as Gaussian, while simultaneously (ii) preserving the stepwise prediction trajectory. However, this hybrid objective can degrade performance and reduce robustness, and it is applicable only to stochastic sampling. Moreover, empirical evidence shows that the independent quantization residual $\Delta'\epsilon_\theta$ has a large deviation from a Gaussian distribution. Directly using it may undermine the theoretical assumptions of the method.

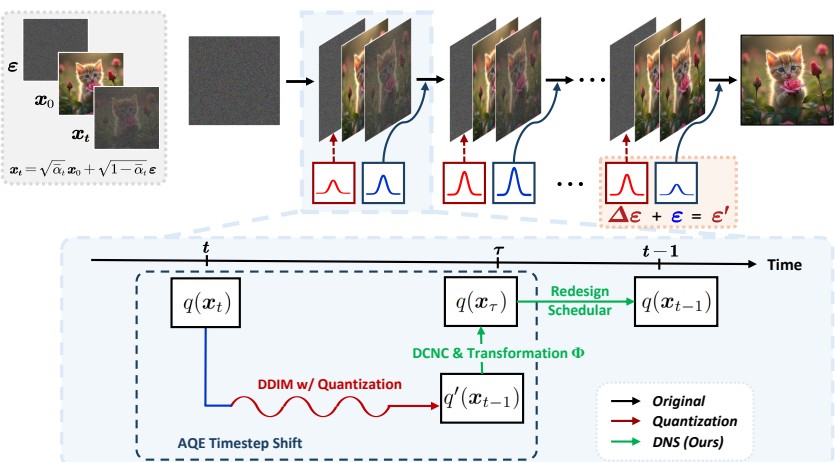

Figure 3: Overview of Deformable Noise Scheduler (DNS). (a) Top: Quantization perturbs the noise prediction (red), shifting each step's marginal away from the full-precision distribution (blue), yielding $\epsilon' = \epsilon + \Delta\epsilon$. These quantization errors are absorbed into the diffusion process, preserving the original prediction distribution. (b) Bottom: DNS interprets effect as *timestep shift*. The quantized distribution $q'(\boldsymbol{x}_{t-1})$ is mapped to original distribution $q(\boldsymbol{x}_\tau)$ via DCNC and transformation $\Phi$. Redesigning scheduler sets a new $\alpha$ so that $\tau$ aligns with $t-1$, maintain original diffusion timesteps.

## 4 METHOD

To address these limitations, we develop a theoretical framework that analyzes how quantization perturbs a diffusion model's predictive distribution and derives distribution-preserving corrections for both the noise statistics and the time parameterization. In Sec. 4.1, we show that the uncorrelated component of the quantization error is heavy-tailed (exhibits excess kurtosis relative to Gaussian noise) and mitigate this mismatch by adding a calibrated uniform correction with lower kurtosis. In Sec. 4.2, we prove that the distribution induced by a quantized model can be mapped to an equivalent non-quantized distribution through an appropriate timestep shift (Fig. 3), enabling the approach to be applied to any generative model with conditionally Gaussian marginals. A more detailed derivation is provided in Appendix A.2.

### 4.1 GAUSSIAN APPROXIMATION OF QUANTIZATION NOISE

When the model is quantized, parameter perturbations introduce errors in the predicted noise. The output of the quantized model can be expressed as:

$$\epsilon'_\theta(\boldsymbol{x}_t, t) = \epsilon_\theta(\boldsymbol{x}_t, t) + \Delta\epsilon_\theta(\boldsymbol{x}_t, t) \tag{11}$$

where $\Delta\epsilon_\theta(\boldsymbol{x}_t, t)$ is a quantization-induced error.

Prior work (He et al., 2023) applied a disentangling transformation (Eq. 8) to separate $\Delta\epsilon_\theta(\boldsymbol{x}_t, t)$ into an "independent" component $\Delta'\epsilon_\theta(\boldsymbol{x}_t, t)$, which was then approximated as Gaussian noise and absorbed into the stochastic sampling term. However, our empirical analysis (see Fig. 4) shows that the residual noise obtained in this manner exhibits excess kurtosis relative to a true Gaussian distribution, violating this assumption.

To correct this, we introduce a refined transformation $T$ such that

$$T(\epsilon'_\theta(\boldsymbol{x}_t, t)) = \epsilon_\theta(\boldsymbol{x}_t, t) + \Delta''\epsilon_\theta(0, \sigma_{\epsilon,t}) \tag{12}$$

where the independent component $\Delta''\epsilon_\theta(0, \sigma_{\epsilon,t})$ is explicitly calibrated to approximate a Gaussian distribution with mean 0 and variance $\sigma_{\epsilon,t}^2$ which depends on the timestep $t$. We refer to this procedure as *Distribution-Calibrated Noise Compensation* (DCNC). DCNC ensures that the residual quantization noise more faithfully matches the Gaussian assumption of the diffusion process, thereby improving stability and generation quality.

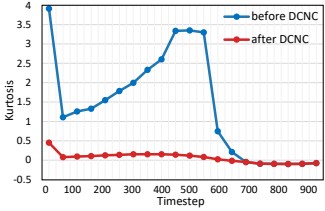

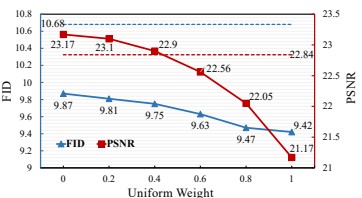

Figure 4: Quantization noise distribution before and after DCNC in timestep 250

Figure 5: Kurtosis of quantization noise before and after DCNC

Figure 6: Performance comparison with different strength of correction

**Distribution-Calibrated Noise Compensation**    To drive the excess kurtosis of the residual toward 0 (i.e., make it behave more like Gaussian noise), we add a zero-mean uniform component. Let the residual have variance $\sigma_o^2$ and excess kurtosis $\kappa_o > 0$. Then the variance of the injected uniform term and the resulting total variance are (derivation in Appendix A.1):

$$\sigma_u^2 = \sigma_o^2 \cdot \sqrt{5\kappa_o/6}\,, \qquad \sigma_n^2 = \sigma_o^2 \cdot \left(1 + \sqrt{5\kappa_o/6}\right) \tag{13}$$

As shown in Fig. 4 and Fig. 5, this uniform compensation substantially reduces the heavy-tailedness of the uncorrelated quantization residual, bringing its excess kurtosis close to the Gaussian value (0). More higher-order moments are also collected in Appendix.A.1 to demonstrate that residuals after DCNC are closer to a Gaussian. Accurate variance estimation is important for applying Eq. 13 and for the marginal analysis in Eq. 17. Because conventional estimators are sensitive to outliers, we use a robust IQR-based estimate $\hat{\sigma}^2 = (\text{IQR}/1.349)^2$, where $\text{IQR} = Q_3 - Q_1$ denotes the interquartile range, with $Q_1$ and $Q_3$ representing the first and third quartiles. Putting this together, our calibrated transformation is:

$$T(\epsilon'_\theta(\boldsymbol{x}_t, t)) = \epsilon'_\theta(\boldsymbol{x}_t, t)/(1 + k) + W_u \cdot U(0, \sigma_u^2) \tag{14}$$

where $k$ (the correlation slope) and $\sigma_u^2$ are estimated during calibration, $U(0, \sigma_u^2)$ denotes a zero-mean uniform random variable with variance $\sigma_u^2$, and $W_u \in [0, 1]$ is a tunable weight that controls the strength of the correction (see Sec. 5.3). Unless stated otherwise, all model outputs are passed through $T(\cdot)$ before subsequent analysis and sampling.

## 4.2    Deformation of Sampling with Quantization Noise

After applying DCNC, the residual quantization error is well-approximated as zero-mean Gaussian. We leverage this property to *absorb* quantization effects into the diffusion process via a *timestep shift*: instead of introduced extra noise, we preserve distributional consistency via a timestep shift $\tau$, aligning the quantized marginal distribution at t with the full-precision marginal distribution at $\tau$. This distribution-perserving perspective removes the reliance on specific stochastic samplers and generally applies across various generative models with Gaussian conditional paths.

**Absorbing Quantization Error by Timestep Shift**    In diffusion models, the reverse process $p_\theta(\boldsymbol{x}_{t-1} \mid \boldsymbol{x}_t)$ reconstructs $\boldsymbol{x}_{t-1}$ from the noisy sample $\boldsymbol{x}_t$ by leveraging an estimate $\hat{\boldsymbol{x}}_0$ of the clean data $\boldsymbol{x}_0$, as formalized in Eq. 4. When the model is quantized, the prediction $\hat{\boldsymbol{x}}_0$ is perturbed by quantization noise (Eq. 11), after correction (Eq. 12), yielding a modified estimate $\hat{\boldsymbol{x}}'_0$. Consequently, the reverse process distribution under quantization becomes:

$$p'_\theta(\boldsymbol{x}_{t-1} \mid \boldsymbol{x}_t) = q_\sigma(\boldsymbol{x}_{t-1} \mid \boldsymbol{x}_t, \hat{\boldsymbol{x}}'_0) = \mathcal{N}\big(\boldsymbol{x}_{t-1}; \hat{\mu}_t(\boldsymbol{x}_t, \hat{\boldsymbol{x}}_0), \big(\sigma_t^2 + C_1^2 \sigma_{\epsilon,t}^2\big)\mathbf{I}\big), \tag{15}$$

where

$$C_1 = \sqrt{\bar{\alpha}_{t-1}} - \sqrt{\frac{(1 - \bar{\alpha}_{t-1} - \sigma_t^2)\bar{\alpha}_t}{1 - \bar{\alpha}_t}}, \tag{16}$$

and $\sigma_{\epsilon,t}^2$ denotes the variance of the independent quantization residual at timestep $t$. Notably, quantization noise does not alter the mean of the reverse distribution but inflates its variance by an additive term $C_1^2 \sigma_{\epsilon,t}^2$, reflecting the uncertainty introduced by model quantization.

Thus the induced conditional marginal distribution under quantization is

$$q'(\boldsymbol{x}_{t-1}|\boldsymbol{x}_0) = \mathcal{N}\big(\boldsymbol{x}_{t-1}; \sqrt{\bar{\alpha}_{t-1}}\,\boldsymbol{x}_0, (1 - \bar{\alpha}_{t-1} + C_1^2\sigma_{\epsilon,t}^2)\mathbf{I}\big). \tag{17}$$

We see that the mean matches the original forward process, but the variance includes an additional quantization-dependent term $C_1^2\sigma_{\epsilon,t}^2$. As a result, quantized marginal distribution $q'(\boldsymbol{x}_{t-1}|\boldsymbol{x}_0)$ no longer coincides with the forward marginal of any discrete timestep in the original diffusion schedule.

To restore equivalence, we introduce a scaling factor $C_2$ and a shifted timestep $\tau > t - 1$ such that:

$$q(C_2\boldsymbol{x}_\tau|\boldsymbol{x}_0) = \mathcal{N}\big(C_2\boldsymbol{x}_\tau; C_2\sqrt{\bar{\alpha}_\tau}\boldsymbol{x}_0, C_2^2(1 - \bar{\alpha}_\tau)\mathbf{I}\big). \tag{18}$$

It can be shown that setting $C_2 = \sqrt{1 + C_1^2\sigma_{\epsilon,t}^2}$ and $\bar{\alpha}_\tau = \bar{\alpha}_{t-1}/C_2^2$ makes $q'(\boldsymbol{x}_{t-1}|\boldsymbol{x}_0)$ and $q(C_2\boldsymbol{x}_\tau|\boldsymbol{x}_0)$ statistically equivalent. This establishes quantization as a *timestep shift* in diffusion process.

**Deformable Noise Scheduler**    Building on this insight, we introduce the *Deformable Noise Scheduler (DNS)*, which maps quantized diffusion states to adjusted timesteps in the full-precision schedule. From Eq. 18, the mapping $\Phi$ is:

$$\boldsymbol{x}_\tau = \Phi(\boldsymbol{x}_{t-1}) = \boldsymbol{x}_{t-1}/C_2. \tag{19}$$

Direct application, however, faces two challenges: (i) $\tau$ is typically non-integer, complicating integration into discrete samplers, and (ii) rounding $\tau$ introduces additional error and potentially extra denoising steps.

To avoid this, we redesign the scheduler by redefining the variance schedule at the previous timestep in the quantized process as $\bar{\alpha}_{t-1}^q$. From Eq. 18, we obtain:

$$\bar{\alpha}_\tau = \frac{\bar{\alpha}_{t-1}^q}{1 + C_1^2\sigma_{\epsilon,t}^2}. \tag{20}$$

Solving Eq. 20 for $\bar{\alpha}_{t-1}^q$ aligns $\tau$ with $t - 1$, ensuring that quantized updates maintain consistent with the original diffusion timesteps.

DNS thus dynamically adapts the noise schedule according to $\sigma_{\epsilon,t}^2$, preserving distributional equivalence without introducing additional steps or overhead. By the same reasoning, DNS naturally extends to flow-matching with Gaussian conditional paths (see Appendix A.3).

## 5 EXPERIMENTS

### 5.1 SETTINGS

**Models and Datasets** We benchmark our method using LDM-4 (Rombach et al., 2022), DiT-XL (Peebles & Xie, 2023), and FLUX.1 (Labs, 2024) (Flow-matching). Following previous works (Li et al., 2025), we randomly sample 3,000 text prompts from COCO Captions 2014 (Chen et al., 2015) for calibration. This calibration set is used to determine the quantization parameters and estimate noise characteristics across all models. For our evaluation datasets, we use ImageNet for class-conditional generation. For text-to-image generation, we use the MJHQ-30K dataset (Li et al., 2024). Our experiments were conducted on a NVIDIA A100-40G GPU. However, it should be noted that our method does not require additional memory usage, as the main storage cost is about network parameters. Both LDM and DIT-XL can run on smaller GPUs.

**Baselines** We compare our method with several PTQ techniques. (1) LDM-4: Tested with 20 steps, guidance scale 3.0, and 256×256 resolution. We compare the FP16 model, Q-Diffusion (W4A8) (Li et al., 2023), and its combination with PTQD (He et al., 2023). (2) DiT-XL: Evaluated with 20 steps, guidance scale 1.5, and 256×256 resolution. We compare the FP16 model, PTQ4DiT (W8A8, W4A8) (Wu et al., 2024), and its combination with PTQD. (3) FLUX.1: Tested with 20 steps, guidance scale 3.5, and 256×256 resolution. Since original SVDQuant have W4A4 Transformer and W16A16 LoRA without complete quantization, we further quantized the LoRA branch of SVDQuant to 6-bit weight. We compare the BP16 model and SVDQuant (W4A4) with W16A16 and W6A16 LoRA (Li et al., 2025). PTQD does not support flow-matching models.

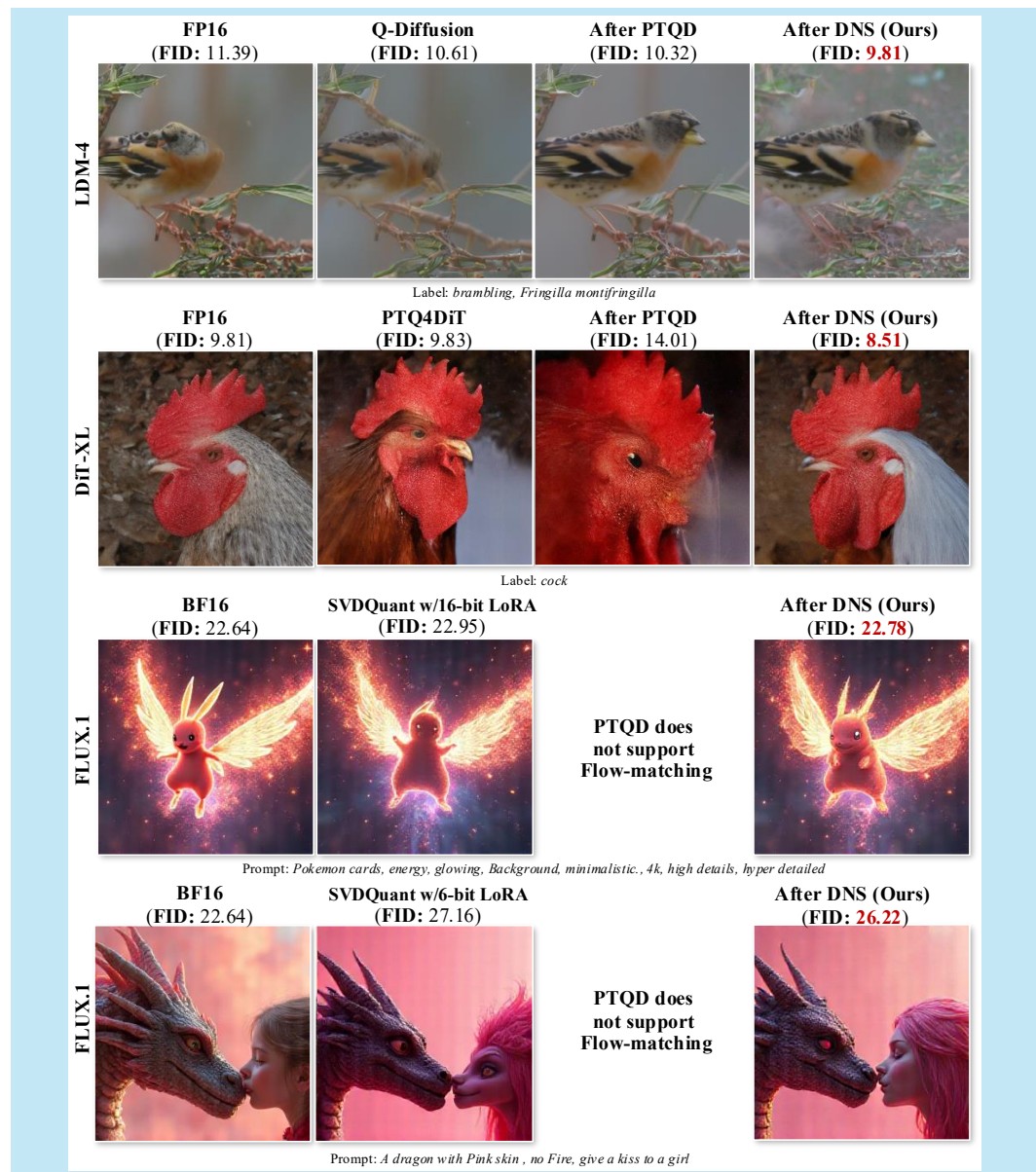

Figure 7: We tested the results on several models (including diffusion and Flow-matching) and compared the results of our method with FP, PTQ, PTQ+PTQD. Numbers in brackets are FID↓.

**Metrics** Following previous works (Li et al., 2025), we evaluate generation quality using several metrics. Fréchet Inception Distance (FID (Heusel et al., 2017; Parmar et al., 2022)) and sliding Fréchet Inception Distance (sFID (Ding et al., 2022)) are particularly well-suited for reflecting distribution similarity, as it measures the distance between feature distributions extracted from generated and real data using a pre-trained neural network. Inception Score (IS (Salimans et al., 2016)) aims to assess quality and diversity. CLIPScore (CLIP (Hessel et al., 2021)) aims to measure text-image alignment. Image Reward (IR (Xu et al., 2023)) aims to approximate human judgment. We also employ Peak Signal Noise Ratio (PSNR) to measures pixel-level trajectory difference, but this metric cannot validate whether the final outputs achieve better distribution preservation or generation quality. Note that in class-conditional generation we use the tag name with scientific nomenclature to calculate IR.

Table 1: Performance comparison across different models

| Model | Bit | Method | Evaluation Metrics | | | | | |
| | | | FID↓ | sFID↓ | IS↑ | CLIP↑ | IR↑ | PSNR↑ |
|---|---|---|---|---|---|---|---|---|
| LDM-4 | FP16 | - | 11.49 | 9.99 | 104.74 | 31.91 | 0.134 | - |
| | W4A8 | Q-Diffusion | 10.68 | 13.71 | 101.95 | 31.97 | -0.014 | 22.84 |
| | | w/PTQD | 10.32 | 12.39 | 100.84 | 31.89 | -0.031 | 23.13 |
| | | w/ours | **9.81** | **9.42** | **102.40** | 31.91 | **0.015** | 23.10 |
| DiT-XL | FP16 | - | 9.81 | 21.11 | 76.09 | 29.84 | -0.349 | - |
| | W8A8 | PTQ4DiT | 9.02 | 18.42 | 76.18 | 29.85 | -0.345 | 26.87 |
| | | w/PTQD | 8.03 | 16.90 | 78.66 | 30.04 | -0.338 | 26.04 |
| | | w/ours | **8.00** | **16.81** | **78.98** | 30.00 | -0.342 | 26.43 |
| | W4A8 | PTQ4DiT | 9.83 | 17.52 | 70.79 | 29.92 | -0.508 | 17.42 |
| | | w/PTQD | 14.01 | 17.13 | 64.67 | 29.73 | -0.609 | 16.32 |
| | | w/ours | **8.51** | **16.64** | **72.65** | **30.01** | **-0.484** | **17.43** |
| FLUX.1 | BF16 | - | 22.64 | 54.23 | 24.15 | 29.30 | 0.900 | - |
| | W4A4 w/W16 LoRA | SVDQuant | 22.95 | 53.64 | 23.80 | 29.21 | 0.877 | 22.21 |
| | | w/ours | **22.78** | **53.15** | **24.01** | 29.21 | **0.890** | 21.92 |
| | W4A4 w/W6 LoRA | SVDQuant | 27.16 | 58.15 | 22.68 | 28.89 | 0.693 | 16.17 |
| | | w/ours | **26.22** | **57.36** | **23.58** | **29.01** | **0.709** | 15.87 |

## 5.2 Main Results

We present our main results in Tab.1 and Fig.7. For class-conditional experiments we generate 30,000 images, and for text-to-image (MJHQ) we generate 10,000 images for evaluation. Observing our experimental results, we find substantial improvements in generation quality metrics including FID (8.1%-13.4% improvement), IS, and IR, demonstrating that our method effectively preserves the original data distribution. Although PSNR decreases slightly in some settings (0.1%-1.6%), CLIP scores remain essentially unchanged ($\leqslant 0.2\%$), confirming that semantic alignment with text conditions is preserved. Since CLIP scores confirm semantic consistency, the PSNR differences reflect different high-quality sampling paths under identical semantic constraints, not semantic drift(worse CLIP) or quality loss(worse FID).

Specifically, for LDM-4 quantized to W4A8 with Q-Diffusion, our approach outperforms both the baseline quantized model and PTQD, improving FID from 10.68 to 9.81 and IR from -0.014 to 0.015. For DiT-XL with PTQ4DiT at W8A8 and W4A8, PTQD is slightly stronger at W8A8 but exhibits marked instability at W4A8 and fails to produce acceptable results; in contrast, our method is robust at both precisions, improving FID from 9.02 to 8.20 (W8A8) and from 9.83 to 8.51 (W4A8). For FLUX.1 under flow matching with SVDQuant at W4A4 with full precision LoRA, even where the baseline quantization method performs exceptionally close to full precision, our method applies directly and further improves results, reducing FID from 22.95 to 22.78 and increasing IR from 0.877 to 0.890. For its with 6-bit LoRA, all weights required for network inference are 6 bits or less, LoRA branch quantized model increases FID from 22.64 to 27.16. After applying our DNS method decrease FID from 27.16 to 26.22. More visualizations are provided in Appendix A.4.

## 5.3 Ablation

**Ablation of Optimization Methods** As shown in Tab. 3, we conduct a systematic ablation study to evaluate the contribution of each component in our framework. Starting from the W4A8 baseline quantized model (with Q-Diffusion) and applying CNC preprocessing, we progressively incorporate our proposed techniques to demonstrate their effectiveness. First, the DNS absorbs quantization noise through a timestep shift, adjusting the diffusion schedule to preserve the original denoising

Table 2: Performance comparison of different variance estimation methods

| Model | Method | FID↓ | IR↑ | PSNR↑ |
|---|---|---|---|---|
| LDM-4 | - | 9.85 | 0.012 | 23.09 |
| | MAD | 9.83 | 0.014 | 23.10 |
| | IQR | 9.81 | 0.015 | 23.10 |
| | KUS | 9.85 | 0.016 | 23.12 |
| | KDE | 9.83 | 0.012 | 23.10 |

Table 3: Effect of different components

| Model | Method | FID↓ | IR↑ | PSNR↑ |
|---|---|---|---|---|
| LDM-4 | base | 10.61 | -0.01 | 22.81 |
| | +CNC | 10.32 | -0.031 | 23.13 |
| | +DNS | 10.02 | 0.009 | 22.83 |
| | +DCNC | 9.81 | 0.015 | 23.10 |

distribution. Second, the DCNC refines noise estimation by more accurately modeling the statistical properties of quantization errors.

**Variance Estimation** As shown in Tab. 2, we test five statistical methods for variance estimation: standard deviation, mean absolute deviation (MAD), interquartile range (IQR), kurtosis-based correction (KUS), and kernel density estimation (KDE). It can be seen that IQR demonstrates best performance on multiple metrics simultaneously.

**Uniform Weight** Fig. 6 shows our experiments with different strength of correction ($W_u$) ranging from 0 to 1. We observed an interesting trade-off: higher $W_u$ values improve generation quality (FID) but reduce similarity to full-precision outputs (PSNR). We select $W_u = 0.2$ for our experiments, as this value achieves a favorable balance between maintaining reasonable similarity while still improving the generation quality.

More ablation experiments are provided in Appendix A.5.

## 6 CONCLUSION

In this paper, we have introduced a novel quantization error absorbing method for diffusion models that effectively mitigates quality degradation after quantization. Unlike previous method, our method is distribution-preserving: it explicitly aligns the quantized model's process distribution with the unquantized diffusion distribution. And our solution works across stochastic, deterministic, and Flow-matching frameworks with Gaussian conditional probability paths. Extensive experiments validate that our method significantly improves generation quality in quantized diffusion models while maintaining computational efficiency.

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

## A APPENDIX

### A.1 DETAILS OF KURTOSIS-BASED UNIFORM NOISE COMPENSATION

Kurtosis measures the "tailedness" of a probability distribution, indicating whether data has heavy tails (more outliers) or light tails compared to a normal distribution. For a random variable $x$ with mean $\mu$, its kurtosis is defined as:

$$\kappa_x = \frac{\mathbb{E}[(X - \mu)^4]}{(\mathbb{E}[(X - \mu)^2])^2} - 3 \tag{21}$$

A positive kurtosis suggests heavier tails and a sharper peak, while a negative kurtosis indicates lighter tails and a flatter distribution. For instance, a Gaussian variable has a kurtosis of 0 while that of a uniform variable is -1.2.

To compensate for the positive kurtosis $\kappa_o$ of residual noise with variance $\sigma_o^2$, we introduce uniform noise with variance $\sigma_u^2$ and kurtosis $\kappa_u < 0$. Given that the two are independent, the kurtosis of their summation can be easily obtained:

$$\kappa_{new} = \frac{\kappa_o \cdot \sigma_o^4 + \kappa_u \cdot \sigma_u^4}{(\sigma_o^2 + \sigma_u^2)^2} \tag{22}$$

Table 4: Statistical moments of quantization noise before and after DCNC

| Timestep | Skewness | $\kappa_x$ | $\kappa_x$ w/DCNC | $\mu_5$ | $\kappa_6$ | $\kappa_6$ w/DCNC |
|---|---|---|---|---|---|---|
| 950 | $0.26\times10^{-2}$ | -0.07 | -0.07 | $1.03\times10^{-8}$ | -0.47 | -0.47 |
| 850 | $0.24\times10^{-2}$ | -0.01 | -0.10 | $0.03\times10^{-8}$ | -0.85 | -0.85 |
| 750 | $0.07\times10^{-2}$ | -0.09 | -0.09 | $0.39\times10^{-8}$ | -0.40 | -0.40 |
| 650 | $-0.51\times10^{-2}$ | 0.21 | -0.02 | $-9.29\times10^{-8}$ | 9.18 | 0.81 |
| 550 | $-0.94\times10^{-2}$ | 3.30 | 0.08 | $-0.03\times10^{-4}$ | 81.38 | 6.28 |
| 450 | $-0.88\times10^{-2}$ | 3.34 | 0.14 | $-0.16\times10^{-4}$ | 145.49 | 9.45 |
| 350 | $-0.62\times10^{-2}$ | 2.33 | 0.15 | $-0.28\times10^{-4}$ | 138.41 | 9.75 |
| 250 | $-0.39\times10^{-2}$ | 1.78 | 0.14 | $-0.52\times10^{-4}$ | 101.13 | 8.82 |
| 150 | $-0.36\times10^{-2}$ | 1.33 | 0.10 | $-1.59\times10^{-4}$ | 68.45 | 7.23 |
| 50 | $-0.51\times10^{-2}$ | 1.11 | 0.08 | $-9.26\times10^{-4}$ | 48.54 | 5.82 |

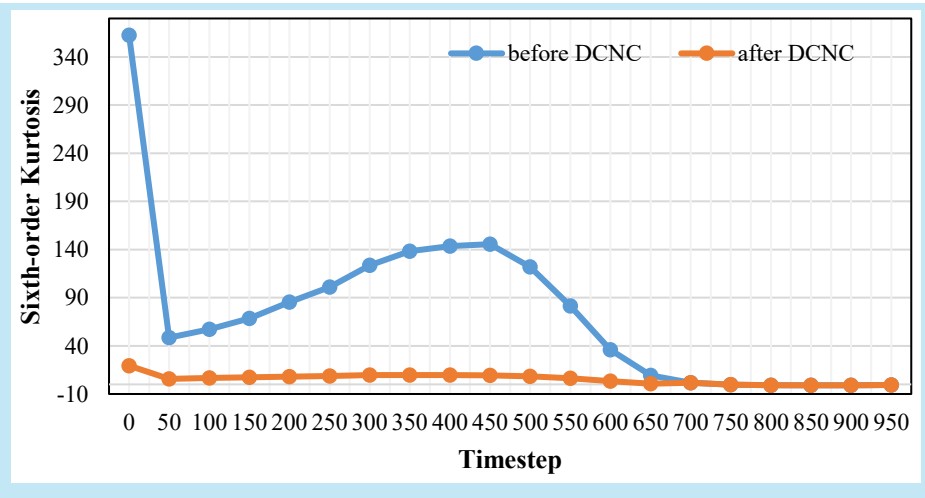

Figure 8: Sixth-order Kurtosis before and after DCNC.

Setting $\kappa_{new}$ to zero and substituting $\kappa_u = -1.2$, we have:

$$\sigma_u^2 = \sigma_o^2 \cdot \sqrt{\frac{5\kappa_o}{6}} \tag{23}$$

Consequently, the compensated noise has a variance of $\sigma_{new}^2 = \sigma_o^2 \cdot \left(1 + \sqrt{\frac{5\kappa_o}{6}}\right)$ and a kurtosis of 0, which can be regarded as a Gaussian distribution.

The uniform distribution is an attractive choice for DCNC because it's kurtosis is very low(excess kurtosis -1.2) and can be sampled efficiently on standard hardware. Unless otherwise stated, all experiments in paper use uniform distribution as the compensation distribution in DCNC.

**More Statistical Analysis** In order to more clearly demonstrate that the compensated residual distribution is closer to a Gaussian, we further measure skewness and higher-order central moments of the residuals. excess kurtosis. Based on the definition of kurtosis, we also define a sixth-order excess kurtosis $\kappa_6 = \frac{\mu_6}{\sigma^6} - 15$ which equals 0 for a Gaussian distribution. We collect the outputs of the LDM-4 model with Q-Diffusion quantized to W4A8 and compute the above statistics over residuals. Results for part of representative timesteps are shown in Tab. 4.

We further visualize the comparison of Kurtosis and Sixth-order Kurtosis before and after DCNC in Fig. 4 and Fig. 8. We also summarize statistics over all timesteps with error bars in Tab. 5.

In these tables, the skewness and fifth-order central moment are collected after compensation, and sixth-order excess kurtosis $\kappa_6$ is collected both before and after compensation for comparison. The

Table 5: Statistics over all timesteps with error bars.

| Skewness ($\times 10^{-3}$) | $\kappa_x$ | $\kappa_x$ w/DCNC | $\mu_5$ ($\times 10^{-4}$) | $\kappa_6$ | $\kappa_6$ w/DCNC |
|---|---|---|---|---|---|
| $-3.65 \pm 4.15$ | $1.42 \pm 1.38$ | $0.02 \pm 0.12$ | $-1.30 \pm 2.85$ | $76.09 \pm 86.71$ | $5.57 \pm 5.25$ |

Table 6: Effect of different compensation distributions in DCNC

| DCNC distribution | FID↓ | sFID↓ | IS↑ |
|---|---|---|---|
| Uniform | 9.65 | 9.33 | 101.96 |
| Triangular | 9.60 | 9.14 | 102.30 |
| Bimodal Gaussian mixture | 9.77 | 9.69 | 101.60 |
| Generalized Gaussian | 9.66 | 9.35 | 102.04 |

results show that, after compensation, skewness and the fifth moment(odd-order moments) are close to zero, indicating that the residuals are nearly symmetric. At the same time, both excess kurtosis and sixth-order excess kurtosis $\kappa_6$ are substantially reduced, providing additional evidence that DCNC makes the residual distribution significantly closer to a Gaussian.

**More Low-kurtosis Distribution Choices** Besides uniform distribution, there exist many other low-kurtosis choices that can be used for compensation. In this work we consider four such noise distribution : uniform distribution $U(-a, a)$ ($\kappa$=-1.2), symmetric triangular distribution on $[-a, a]$ ($\kappa$=-0.6), bimodal Gaussian mixture ($\kappa$=-0.5 with r=1), and generalized Gaussian (exponential-power) distribution(thus $\kappa \approx$-0.81 with $\beta$=4). Ablation results over these distributions are shown in Tab. 6.

From this table, we observe that the triangular distribution achieves better overall performance than uniform distribution, indicating that using a triangular noise distribution for DCNC compensation is also a viable choice. Nevertheless, due to uniform distribution extremely simple and efficient sampling properties on standard hardware, we retain the uniform distribution as the default compensation distribution throughout this paper.

## A.2 EXTENDED DETAILS AND SUPPLEMENTARY MATERIAL FOR THE MAIN DERIVATION

In Sec. 4.1, I derived the transformation function $T$. Next, I will provide a detailed derivation of the transformation function $T$.

From Eq. 8 and Eq. 11, we have:

$$\frac{\epsilon'_\theta(\boldsymbol{x}_t, t)}{1+k} = \epsilon_\theta(\boldsymbol{x}_t, t) + \frac{\Delta' \epsilon_\theta(\boldsymbol{x}_t, t)}{1+k} \tag{24}$$

Following the idea of PTQD, we can treat the latter term as random noise, $\frac{\Delta' \epsilon_\theta(\boldsymbol{x}_t, t)}{1+k} \approx \eta_\theta$. Experiments show that $\eta_\theta$ has higher kurtosis than Gaussian noise. Therefore, we introduce a uniform distribution component. Combined with compensation using Eq. 13:

$$\frac{\Delta' \epsilon_\theta(\boldsymbol{x}_t, t)}{1+k} + U(0, \sigma_u^2) = \Delta'' \epsilon_\theta(0, \sigma_{\epsilon,t}) \sim \mathcal{N}(0, \sigma_{\epsilon,t}^2) \tag{25}$$

In practice, we introduce a factor $W_u$ to control the final proportion added. The analysis of $W_u$ is presented in Sec. 5.3. Thus, we can formulate our improved transformation $T$ as Eq. 14:

$$T(\epsilon'_\theta(\boldsymbol{x}_t, t)) = \frac{\epsilon'_\theta(\boldsymbol{x}_t, t)}{1+k} + W_u \cdot U(0, \sigma_u^2) \tag{26}$$

$$= \epsilon_\theta(\boldsymbol{x}_t, t) + \frac{\Delta' \epsilon_\theta(\boldsymbol{x}_t, t)}{1+k} + W_u \cdot U(0, \sigma_u) \tag{27}$$

$$= \epsilon_\theta(\boldsymbol{x}_t, t) + \Delta''_{\epsilon_\theta}(0, \sigma_{\epsilon,t}) \tag{28}$$

where $\sigma_{\epsilon,t}$ and $k$ are statistically computed from the calibration.

When employing a quantized model, quantization introduces parameter perturbations that manifest as errors in noise prediction. The noise predicted by quantized model can be represented as Eq. 11:

$$\epsilon'_\theta(\boldsymbol{x}_t, t) = \epsilon_\theta(\boldsymbol{x}_t, t) + \Delta_{\epsilon_\theta}(\boldsymbol{x}_t, t) \tag{29}$$

where $\epsilon_\theta(\boldsymbol{x}_t, t)$ represents the original network prediction, $\epsilon'_\theta(\boldsymbol{x}_t, t)$ represents the quantized network prediction, and $\Delta_{\epsilon_\theta}(\boldsymbol{x}_t, t)$ captures the quantization-induced error.

To ensure that the noise distribution introduced by quantization errors approximates a Gaussian distribution, we apply a transformation $T$ to this error function Eq. 12:

$$T(\epsilon'_\theta(\boldsymbol{x}_t, t)) = \epsilon_\theta(\boldsymbol{x}_t, t) + \Delta''_{\epsilon_\theta}(0, \sigma_{\epsilon,t})$$

where $\Delta''_{\epsilon_\theta}(0, \sigma_{\epsilon,t})$ follows a Gaussian distribution with mean 0 and variance $\sigma_{\epsilon,t}^2$ that depends on the timestep $t$.

Substituting Eq. 12 into Eq. 3, we obtain:

$$\hat{\boldsymbol{x}}'_0 = \frac{1}{\sqrt{\bar{\alpha}_t}}(\boldsymbol{x}_t - \sqrt{1 - \bar{\alpha}_t} T(\epsilon'_\theta(\boldsymbol{x}_t, t))) \tag{29}$$

$$= \frac{1}{\sqrt{\bar{\alpha}_t}}(\boldsymbol{x}_t - \sqrt{1 - \bar{\alpha}_t} \epsilon_\theta(\boldsymbol{x}_t, t)) - \frac{\sqrt{1 - \bar{\alpha}_t}}{\sqrt{\bar{\alpha}_t}} \Delta''_{\epsilon_\theta}(0, \sigma_{\epsilon,t}) \tag{30}$$

$$= \hat{\boldsymbol{x}}_0 - \frac{\sqrt{1 - \bar{\alpha}_t}}{\sqrt{\bar{\alpha}_t}} \Delta''_{\epsilon_\theta}(0, \sigma_{\epsilon,t}) \tag{31}$$

where the estimation of $\boldsymbol{x}_0$ using the quantized model, denoted as $\hat{\boldsymbol{x}}_0'$. Consequently, the reverse process under quantization is shown as Eq. 15:

$$p_\theta'(\boldsymbol{x}_{t-1} \mid \boldsymbol{x}_t) = \mathcal{N}\left(\sqrt{\bar{\alpha}_{t-1}}\boldsymbol{x}_0' + \sqrt{1 - \bar{\alpha}_{t-1} - \sigma_t^2} \cdot \frac{\boldsymbol{x}_t - \sqrt{\bar{\alpha}_t}\boldsymbol{x}_0'}{\sqrt{1 - \bar{\alpha}_t}}, \sigma_t^2 I_1\right) \tag{32}$$

$$= \mathcal{N}\left(\sqrt{\bar{\alpha}_{t-1}}\boldsymbol{x}_0 + \sqrt{1 - \bar{\alpha}_{t-1} - \sigma_t^2} \cdot \frac{\boldsymbol{x}_t - \sqrt{\bar{\alpha}_t}\boldsymbol{x}_0}{\sqrt{1 - \bar{\alpha}_t}} + C_1\Delta_{\epsilon_\theta}''(0, \sigma_{\epsilon,t}), \sigma_t^2 I\right) \tag{33}$$

$$= \mathcal{N}\left(\sqrt{\bar{\alpha}_{t-1}}\boldsymbol{x}_0 + \sqrt{1 - \bar{\alpha}_{t-1} - \sigma_t^2} \cdot \frac{\boldsymbol{x}_t - \sqrt{\bar{\alpha}_t}\boldsymbol{x}_0}{\sqrt{1 - \bar{\alpha}_t}}, \sigma_t^2 I + C_1^2\sigma_{\epsilon,t}^2 I\right) \tag{34}$$

When using the DDIM sampler for deterministic sampling, $\sigma_t = 0$.

Then we derive the marginal distribution $q'(\boldsymbol{x}_{t-1}|\boldsymbol{x}_0)$ by integrating over the intermediate variable $\boldsymbol{x}_t$:

$$q'(\boldsymbol{x}_{t-1}|\boldsymbol{x}_0) = \int q_\sigma(\boldsymbol{x}_{t-1}|\boldsymbol{x}_t, \boldsymbol{x}_0) \cdot q(\boldsymbol{x}_t|\boldsymbol{x}_0) \, d\boldsymbol{x}_t \tag{35}$$

Rigorously deriving this integral is quite complex, so we first provide a simplified derivation. We need to know that any linear combination and marginalization of Gaussian distributions remains Gaussian, allowing us to consider the mean and variance of the integral result separately.

Observing Eq. 34, we find that quantization only affects the variance part of $q'(\boldsymbol{x}_{t-1}|\boldsymbol{x}_t, \boldsymbol{x}_0)$, while the mean part remains unchanged.

First is mean analysis: Since the added variance terms have zero mean, by the law of total expectation, the marginal mean remains unchanged:

$$E[q'(\boldsymbol{x}_{t-1}|\boldsymbol{x}_0)] = E[E[q'(\boldsymbol{x}_{t-1}|\boldsymbol{x}_t, \boldsymbol{x}_0)]|\boldsymbol{x}_0] = \sqrt{\bar{\alpha}_{t-1}}\boldsymbol{x}_0 \tag{36}$$

Then is Variance Analysis: We apply the law of total variance:

$$\text{Var}[q'(\boldsymbol{x}_{t-1}|\boldsymbol{x}_0)] = E[\text{Var}[q'(\boldsymbol{x}_{t-1}|\boldsymbol{x}_t, \boldsymbol{x}_0)]|\boldsymbol{x}_0] + \text{Var}[q'(E[\boldsymbol{x}_{t-1}|\boldsymbol{x}_t, \boldsymbol{x}_0])|\boldsymbol{x}_0] \tag{37}$$

since the first term's independent of $\boldsymbol{x}_t$:

$$E[\sigma_t^2 I_1 + C_t^2\sigma_\epsilon^2 I_2|\boldsymbol{x}_0] = (\sigma_t^2 + C_t^2\sigma_\epsilon^2)\mathbf{I} \tag{38}$$

Since the added variance terms have zero mean, the second term remains consistent with the original diffusion process:

$$\text{Var}[E[q'(\boldsymbol{x}_{t-1}|\boldsymbol{x}_t, \boldsymbol{x}_0)]|\boldsymbol{x}_0] = (1 - \bar{\alpha}_{t-1} - \sigma_t^2)\mathbf{I} \tag{39}$$

Therefore, the final integral result is Eq. 17:

$$q'(\boldsymbol{x}_{t-1}|\boldsymbol{x}_0) = \mathcal{N}(\boldsymbol{x}_{t-1}; \sqrt{\bar{\alpha}_{t-1}}\boldsymbol{x}_0, (1 - \bar{\alpha}_{t-1} + C_1^2\sigma_{\epsilon,t}^2)\mathbf{I})$$

Consequently, the quantized model marginal distribution $q'(\boldsymbol{x}_{t-1}|\boldsymbol{x}_0)$ does not directly map the data distribution of the original generative process.

To address this, we seek a transformation that maps it to the original diffusion marginal distribution at other timesteps. We introduce a scaling factor $C_2$ and a new timestep $\tau$ that together establish an equivalence between the marginal distribution of the quantized model and that of the original diffusion model. Specifically, we define Eq. 18:

$$q(C_2\boldsymbol{x}_\tau|\boldsymbol{x}_0) = \mathcal{N}(C_2\boldsymbol{x}_\tau; C_2\sqrt{\bar{\alpha}_\tau}\boldsymbol{x}_0, C_2^2(1 - \bar{\alpha}_\tau)\mathbf{I})$$

Through multi-step denoising, this marginal distribution mapping ensures that the final generated distribution matches the original model's distribution. To make $q(\boldsymbol{x}_{t-1}'|\boldsymbol{x}_0)$ and $q(C_2\boldsymbol{x}_\tau|\boldsymbol{x}_0)$ in Eq. 17 and Eq. 18 statistically equivalent, we set their means and variances equal respectively.

Therefore, we solve the system of equations:

$$\begin{cases} \sqrt{\bar{\alpha}_{t-1}} = C_2\sqrt{\bar{\alpha}_\tau} & \text{(mean equality)} \\ (1 - \bar{\alpha}_{t-1}) + C_1^2\sigma_{\epsilon,t}^2 = C_2^2(1 - \bar{\alpha}_\tau) & \text{(variance equality)} \end{cases} \tag{40}$$

Solving this system yields:

$$C_2 = \sqrt{1 + C_1^2 \sigma_{\epsilon,t}^2} \quad \text{and} \quad \bar{\alpha}_\tau = \frac{\bar{\alpha}_{t-1}}{C_2^2} \tag{41}$$

While PTQD (He et al., 2023) also attempt to absorb quantization error into noise terms, our method provides a principled mathematical framework that explicitly maps the quantized distribution back to the forward diffusion process by timestep shift. The resulting compensation mechanism is theoretically guaranteed to perform effectively across stochastic sampling, deterministic sampling, and Flow-matching with Gaussian conditional probability paths. Through absorbing quantization error by timestep shift, our approach provides a more fundamental understanding of quantization in diffusion models while offering practical advantages in implementation flexibility and sampling robustness.

Based on the above derivations, the implementation is given in Algorithm 1 and Algorithm 2.

---

**Algorithm 1:** Deformable Noise Scheduler Calibration

**Input:** Timesteps sequence $\{(t, t_{\text{prev}})\}$, Calibration set $C = \{\boldsymbol{x}_0\}$
**Output:** Calibrated alpha values $\{\bar{\alpha}_{t-1}^q(t)\}$, slopes $\{k(t)\}$, uniform variances $\{\sigma_u^2(t)\}$

1  Initialize $C_\epsilon \leftarrow \emptyset, C_{\Delta\epsilon} \leftarrow \emptyset$;
2  **for** *each* $(t, previous\ t) \in timesteps$ **do**
3      **for** *each* $\boldsymbol{x}_0 \in C$ **do**
4          $\boldsymbol{x}_t \leftarrow \text{AddNoise}(\boldsymbol{x}_0, t)$;
5          $\epsilon_\theta(\boldsymbol{x}_t) \leftarrow \text{OriginalNetwork}(\boldsymbol{x}_t)$;
6          $\epsilon_\theta'(\boldsymbol{x}_t) \leftarrow \text{QuantizedNetwork}(\boldsymbol{x}_t)$;
7          $\Delta\epsilon_\theta \leftarrow \epsilon_\theta'(\boldsymbol{x}_t) - \epsilon_\theta(\boldsymbol{x}_t)$;
8          $C_\epsilon.\text{append}(\epsilon_\theta(\boldsymbol{x}_t)), C_{\Delta\epsilon}.\text{append}(\Delta\epsilon_\theta)$;
9      $k(t), d(t) \leftarrow \text{LinearRegression}(C_\epsilon, C_{\Delta\epsilon})$;
10     $C_{\Delta'\epsilon} \leftarrow \text{ComputeResiduals}(C_\epsilon, C_{\Delta\epsilon}, k(t), d(t))$;
11     $\sigma_{\epsilon,t,\text{o}}^2 \leftarrow \text{Variance}(C_{\Delta'\epsilon})$;
12     $\kappa_\text{o} \leftarrow \text{Kurtosis}(C_{\Delta'\epsilon})$;
13     $\sigma_u^2(t) \leftarrow \sigma_{\epsilon,t,\text{o}}^2 \times \sqrt{\frac{5\kappa_\text{o}}{6}}\ C_{\Delta''\epsilon} \leftarrow \text{Transformation}T(C_{\Delta'\epsilon}, k(t), W_\text{u}, \sigma_u^2(t))$

    $\sigma_{\epsilon,t}^2 \leftarrow \text{Variance}(C_{\Delta''\epsilon})\ \bar{\alpha}_{t-1}^q \leftarrow \text{NewtonSolve}\left(\bar{\alpha}_{t-1} = \frac{\bar{\alpha}_{t-1}^q}{1 + C_1^2 \sigma_{\epsilon,t}^2}\right) \text{Store } \bar{\alpha}_{t-1}^q(t), k(t),$
    $\sigma_u^2(t)$;

14 **return** $\{\bar{\alpha}_{t-1}^q(t)\}, \{k(t)\}, \{\sigma_u^2(t)\}$

15 **Note:** Linear regression models $\Delta\epsilon_\theta = k(t) \cdot \epsilon_\theta + d(t) + \text{residual}$, but $d(t)$ is usually very close to 0

---

---

**Algorithm 2:** Deformable Noise Scheduler Inference

**Input:** Calibrated parameters $\{\bar{\alpha}_{t-1}^q(t)\}, \{k(t)\}, \{\sigma_u^2(t)\}$, Timesteps sequence $\{(t, \text{previous } t)\}$
**Output:** Generated sample $\boldsymbol{x}_0$

1  Initialize $\boldsymbol{x}_T \sim \mathcal{N}(0, I)$;
2  $\boldsymbol{x}_t \leftarrow \boldsymbol{x}_T$;
3  **for** *each* $(t, \_) \in timesteps$ **do**
4      $\epsilon_\theta'(\boldsymbol{x}_t, t) \leftarrow \text{QuantizedNetwork}(\boldsymbol{x}_\text{current}, t)$;
5      $\epsilon_\theta \leftarrow \text{Transformation}T(\epsilon_\theta'(\boldsymbol{x}_t, t), k(t), W_\text{u}, \sigma_u^2(t))\ \hat{\boldsymbol{x}}_0 \leftarrow \frac{\boldsymbol{x}_t - \sqrt{1 - \bar{\alpha}_t}\epsilon_\theta}{\sqrt{\bar{\alpha}_t}}$
    $\boldsymbol{x}_{t-1} \leftarrow \text{DiffusionStep}(\boldsymbol{x}_t, \hat{\boldsymbol{x}}_0, t, \alpha_{t-1} = \bar{\alpha}_{t-1}^q(t), \sigma_t = 0)\ \boldsymbol{x}_t \leftarrow \boldsymbol{x}_{t-1}$;

6  **return** $\boldsymbol{x}_t$

7  **Note:** Next loop uses $t_{\text{prev}}$ (not $t_{\text{prev}}^q$) as the current timestep

---

A side-by-side derivation comparing our DNS to PTQD is shown in Tab. 7.

Table 7: A side-by-side derivation comparing PTQD and our DNS.

| **PTQD (stochastic-sampling view)** | **DNS (ours, distribution-preserving view)** |
|---|---|
| Both PTQD and our DNS first decompose the quantization error (Eq. 8) as $$\Delta\epsilon_\theta(x_t,t) = k\,\epsilon_\theta(x_t,t) + \Delta'\epsilon_\theta(x_t,t).$$ This yields the following quantized conditional: $$p'_\theta(x_{t-1}\mid x_t) = \mathcal{N}\big(x_{t-1}; \hat\mu_t(x_t,\hat x_0),\, \sigma_t^2 I\big) + \xi_t,$$ $$\xi_t \sim \mathcal{N}\big(0,\, C_1^2\sigma_{\epsilon,t}^2 I\big),$$ where $\xi_t$ is the independent Gaussian component of the quantization noise. Using this decomposition and quantized conditional, PTQD then starts from the standard stochastic sampler (Eq. 5): $$x_{t-1} = \frac{1}{\sqrt{\alpha_t}}\,x_t - \beta_t\frac{1}{\sqrt{1-\bar\alpha_t}}\,\epsilon_\theta(x_t,t) - \sigma_t^0 z,$$ $$z \sim \mathcal{N}(0,I).$$ Substituting the decomposed quantization error into Eq. 5 gives PTQD's stochastic update (Eq. 10): $$x_{t-1} = \frac{1}{\sqrt{\alpha_t}}\,x_t - \beta_t\frac{1}{\sqrt{1-\bar\alpha_t}}\,\epsilon_\theta(x_t,t)$$ $$- \sigma_t z - \beta_t\, c_t\,\Delta'\epsilon_\theta(x_t,t),$$ where $c_t$ is a scalar (a function of $\alpha_t, \bar\alpha_t, k$), and the last two terms $-\sigma_t z - \beta_t c_t\Delta'\epsilon_\theta(x_t,t)$ jointly play the role of the original stochastic noise term $\sigma_t^0 z$ in Eq. 5. Thus $\Delta'\epsilon_\theta$ is treated as an extra stochastic noise term injected into the sampling process. | Both PTQD and our DNS first decompose the quantization error (Eq. 8) as $$\Delta\epsilon_\theta(x_t,t) = k\,\epsilon_\theta(x_t,t) + \Delta'\epsilon_\theta(x_t,t).$$ This yields the following quantized conditional: $$p'_\theta(x_{t-1}\mid x_t) = \mathcal{N}\big(x_{t-1}; \hat\mu_t(x_t,\hat x_0),\, \sigma_t^2 I\big) + \xi_t,$$ $$\xi_t \sim \mathcal{N}\big(0,\, C_1^2\sigma_{\epsilon,t}^2 I\big),$$ where $\xi_t$ is the independent Gaussian component of the quantization noise. Our DNS derivation then starts from the corresponding quantized reverse conditional (Eq. 15), which folds the independent quantization noise into the reverse: $$p'_\theta(x_{t-1}\mid x_t) = q_\sigma\big(x_{t-1}\mid x_t, \hat x_0'\big)$$ $$= \mathcal{N}\Big(x_{t-1}; \hat\mu_t(x_t,\hat x_0),\, \big(\sigma_t^2 + C_1^2\sigma_{\epsilon,t}^2\big)I\Big).$$ Thus the induced conditional marginal distribution under quantization is (Eq. 17): $$q'(x_{t-1}\mid x_0) = \mathcal{N}\Big(x_{t-1}; \sqrt{\bar\alpha_{t-1}}\,x_0,\, \big(1-\bar\alpha_{t-1}+C_1^2\sigma_{\epsilon,t}^2\big)I\Big).$$ DNS then uses Eqs. 18–20 to construct a distributional map that identifies this quantized marginal with an unquantized marginal at a shifted time $\tau$; in particular, Eq. 18 gives $$q'(x_{t-1}\mid x_0) \longleftrightarrow q\big(C_2 x_\tau \mid x_0\big),$$ and Eq. 20 defines the time shift $\tau$ and the corresponding time-shifted scheduler $\bar\alpha_{t-1}^{(q)} := \bar\alpha_\tau$, this explicitly constructs a distribution-preserving map that sends the quantized marginal to an original (unquantized) marginal at the shifted time $\tau$. |

Table 8: Concise comparison between DNS and prior work

| Method | Parameter update? | Arch.changes or limits? | Plug-and-play over other PTQ? | Stoch. | Det. | FM |
|---|---|---|---|---|---|---|
| Q-Diffusion | Yes (Calibration, weight/quant update) | No | Not over SVDQuant | Yes | Yes | Yes |
| PTQ4DiT | Yes (Calibration, weight/quant update) | No Designed for DiT | Only DiT | Yes | Yes | Yes |
| SVDQuant | Yes (Weight update and FP16 finetune) | Add FP16 LoRA (Incomplete quant.) | Not over many model-changing PTQ | Yes | Yes | Yes |
| PTQD | No | No | Almost Yes | Yes | No | No |
| **DNS (Ours)** | No | No | Almost Yes | Yes | Yes | Yes |

We also draw a concise comparison table to better compare our method with prior work, shown in Tab. 8

This highlights that DNS is both plug-and-play over many existing PTQ methods and applicable to stochastic(Stoch.), deterministic(Det.), and flow-matching(FM) samplers without architectural/weight changes or additional training.

A.3    DETAILS OF FLOW-MATCHING DERIVATION

In FLUX.1, the forward process is defined as:

$$\boldsymbol{x}_t = (1 - \sigma_t) \cdot \boldsymbol{x}_0 + \sigma_t \cdot \varepsilon \tag{42}$$

where $\varepsilon \sim \mathcal{N}(0, \mathbf{I})$ and $\sigma_t$ represents the noise level at time $t$.

The denoising process follows a discrete update rule:

$$\boldsymbol{x}_{t_{next}} = \boldsymbol{x}_t + \Delta\sigma \cdot v_\theta(\boldsymbol{x}_t, \sigma_t) \tag{43}$$

where $\Delta\sigma = \sigma_{t_{next}} - \sigma_t$ and $v_\theta(\boldsymbol{x}_t, \sigma_t)$ is the velocity field predicted by the model.

When employing a quantized model, the velocity field prediction contains quantization-induced errors:

$$v'_\theta(\boldsymbol{x}_t, \sigma_t) = v_\theta(\boldsymbol{x}_t, \sigma_t) + \Delta v_\theta(\boldsymbol{x}_t, \sigma_t) \tag{44}$$

To ensure that this error approximates a Gaussian distribution, we apply transformation $T$:

$$T(v'_\theta(\boldsymbol{x}_t, \sigma_t)) = v_\theta(\boldsymbol{x}_t, \sigma_t) + \Delta' v_\theta(0, \sigma_{v,t}) \tag{45}$$

where $\Delta' v_\theta(0, \sigma_{v,t}) \sim \mathcal{N}(0, \sigma_{v,t}^2 \mathbf{I})$.

The denoising process with quantization errors becomes:

$$\begin{aligned}
\boldsymbol{x}'_{t_{next}} &= \boldsymbol{x}_t + \Delta\sigma \cdot [v_\theta(\boldsymbol{x}_t, \sigma_t) + \Delta' v_\theta(0, \sigma_{v,t})] \\
&= \boldsymbol{x}_{t_{next}} + \Delta\sigma \cdot \Delta' v_\theta(0, \sigma_{v,t})
\end{aligned} \tag{46}$$

Substituting the forward process expression for $\boldsymbol{x}_t$:

$$\boldsymbol{x}'_{t_{next}} = (1 - \sigma_{t_{next}}) \cdot \boldsymbol{x}_0 + \sigma_{t_{next}} \cdot \varepsilon + \Delta\sigma \cdot \Delta' v_\theta(0, \sigma_{v,t}) \tag{47}$$

Drawing from our diffusion derivation, we seek a transformation that maps $\boldsymbol{x}'_{t_{next}}$ back to the standard forward path. We introduce a scaling factor $C_2$ and a new timestep $\tau$ that together establish an equivalence between the quantized output and a point on the original path. Specifically, we define:

$$C_2 \cdot \boldsymbol{x}_\tau = C_2 \cdot (1 - \sigma_\tau)\boldsymbol{x}_0 + C_2 \cdot \sigma_\tau \varepsilon' \tag{48}$$

Then we can establish that when $C_2 = (1 - \sigma_{t_{next}}) + \sqrt{\sigma_{t_{next}}^2 + \Delta\sigma^2 \sigma_{v,t}^2}$ and $\sigma_\tau = (C_2 + \sigma_{t_{next}} - 1)/C_2$, $\boldsymbol{x}'_{t_{next}}$ and $C_2 \boldsymbol{x}_\tau$ become statistically equivalent, with matching coefficients for $\boldsymbol{x}_0$ and equivalent variance terms for the noise components $\sigma_{t_{next}} \cdot \varepsilon + \Delta\sigma \cdot \Delta' v_\theta(0, \sigma_{v,t})$.

Importantly, neural networks trained for flow-matching are designed to estimate the velocity field pointing from any noisy sample toward the clean data. Since the network is trained to predict $v_\theta(\boldsymbol{x}_t, \sigma_t) \approx \mathbb{E}[\varepsilon | \boldsymbol{x}_t, \sigma_t]$ for any valid noise distribution, it can still function effectively when the noise distribution changes from $\varepsilon$ to $\varepsilon'$. The model will naturally predict the velocity field that directs $\boldsymbol{x}_\tau$ toward $\boldsymbol{x}_0$, allowing our transformation to effectively absorb quantization errors.

## A.4 VISUALIZATION RESULTS

In this section, we demonstrate the visualisation results by different models. (1) LDM-4: Tested with 20 steps, guidance scale 3.0, and 256×256 resolution. We compare the FP16 model, Q-Diffusion (W4A8), its combination with PTQD and its combination with ours. The results are shown in Fig. 9. (2) DiT-XL: Evaluated with 20 steps, guidance scale 1.5, and 256×256 resolution. We compare the FP16 model, PTQ4DiT (W4A8), its combination with PTQD and its combination with ours. The results are shown in Fig. 10.

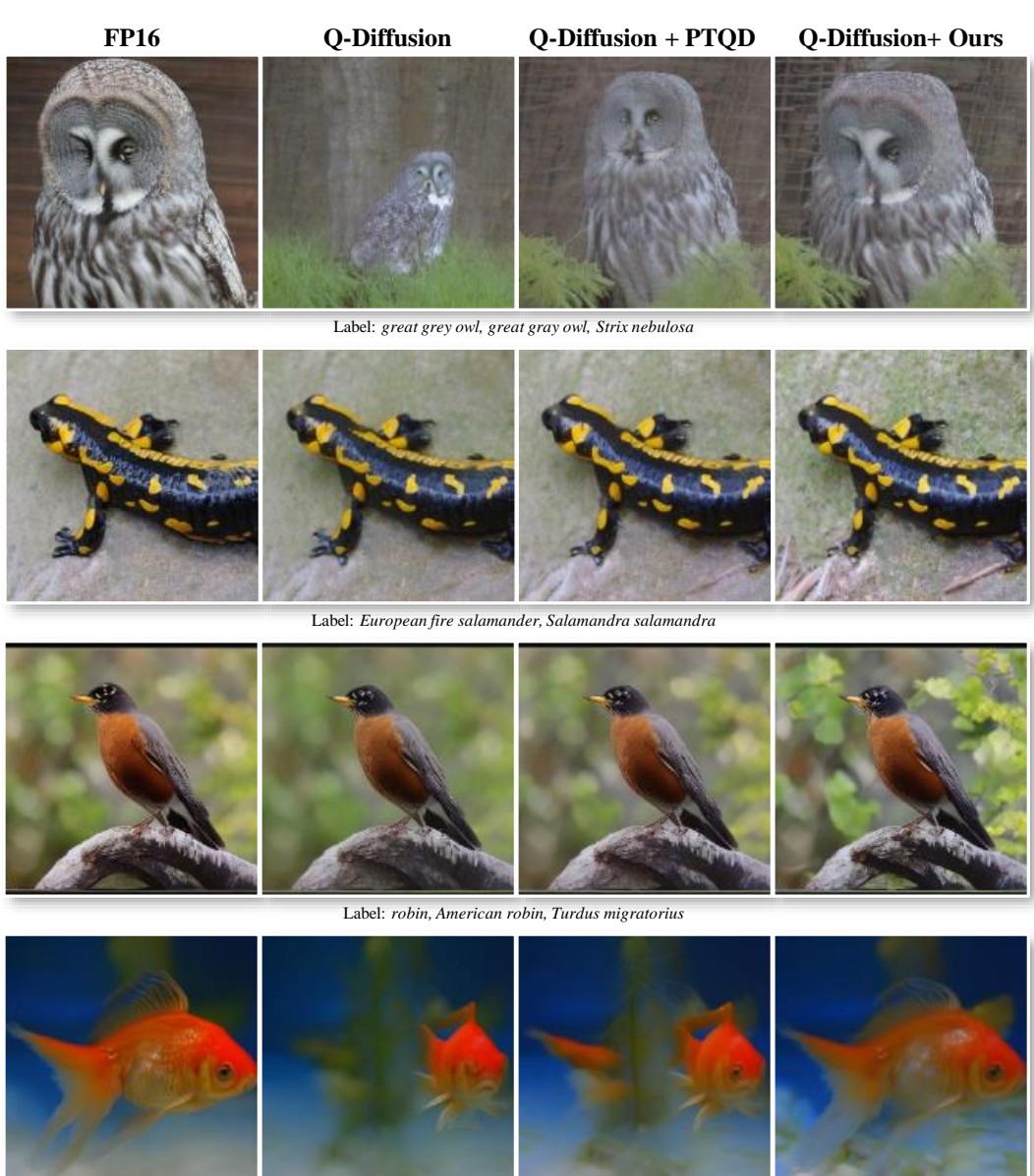

Figure 9: Tested the LDM-4 model with Q-Diffusion quantization to W4A8 precision. The resolution of the generated image is 256

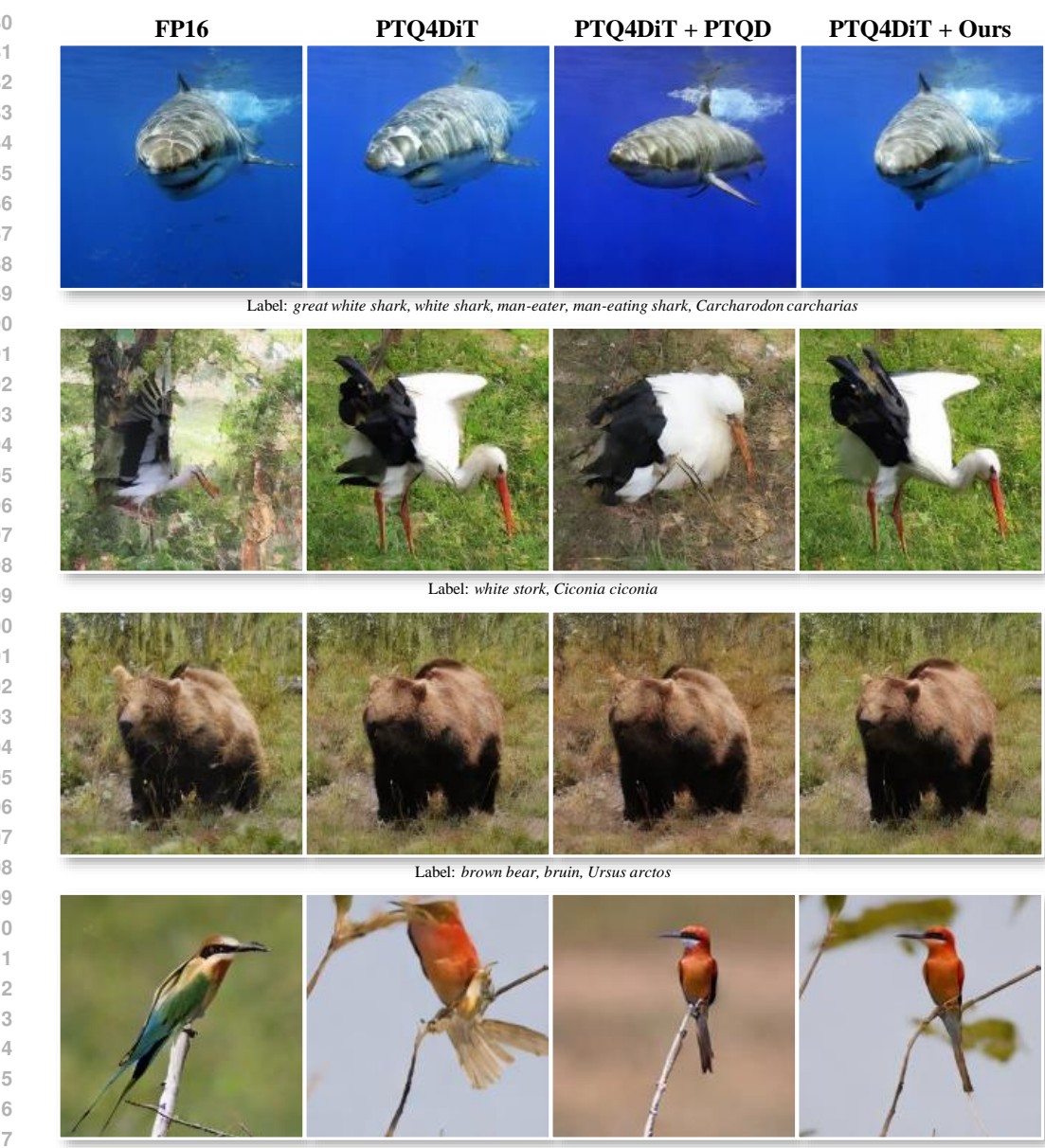

| FP16 | PTQ4DiT | PTQ4DiT + PTQD | PTQ4DiT + Ours |
|------|---------|----------------|----------------|

Label: *great white shark, white shark, man-eater, man-eating shark, Carcharodon carcharias*

Label: *white stork, Ciconia ciconia*

Label: *brown bear, bruin, Ursus arctos*

Label: *bee eater*

Figure 10: Tested the DIT-XL model with PTQ4DiT quantization to W4A8 precision. The resolution of the generated image is 256

(3) FLUX.1 : Tested with guidance scale 3.5, 20 steps at 256×256 and 1024×1024. We compare the FP16 model, SVDQuant (W4A4) with quantized its LoRA branch weight to 6-bit and its combination with ours. PTQD does not support Flow-matching. The results are shown in Fig. 11, Fig. 12, Fig. 13. Fig. 14 and Fig. 15. It can be seen that our method has good results in maintaining semantics, reducing artifacts, increasing details, and improving diversity, which shows that we have successfully achieved the goal of distribution preservation.

(4) FLUX.1 : We also try optimizing SVDQuant (W4A4) with FP16 LoRA branch. Tested with guidance scale 3.5, 20 steps at 256×256 and 1024×1024. We compare FP16 model, SVDQuant (W4A4) with FP16 LoRA and its combination with ours. The results are shown in Fig. 16 and Fig. 17. Our method still has capability to improve under strong baseline.

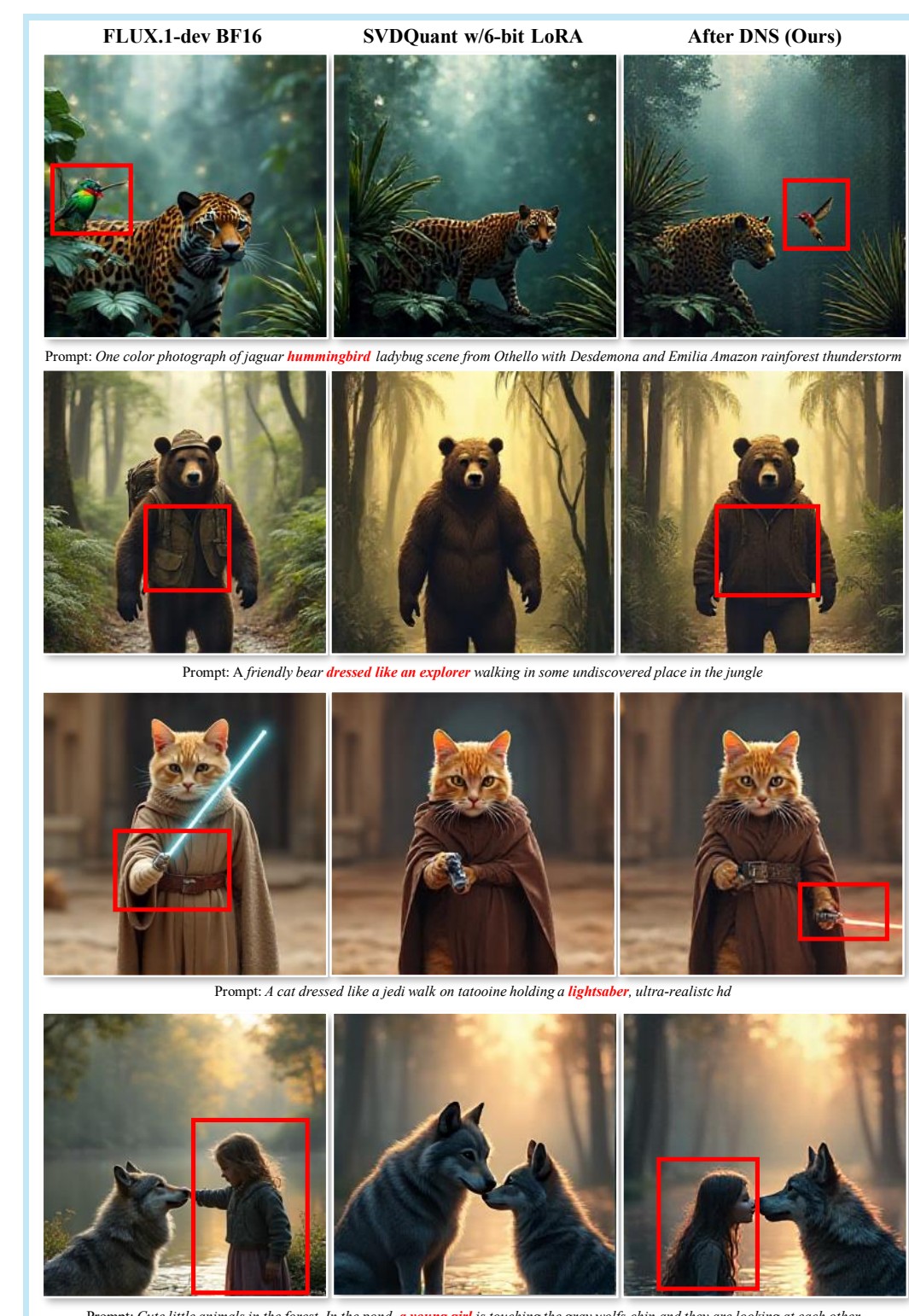

| FLUX.1-dev BF16 | SVDQuant w/6-bit LoRA | After DNS (Ours) |

Prompt: *One color photograph of jaguar **hummingbird** ladybug scene from Othello with Desdemona and Emilia Amazon rainforest thunderstorm*

Prompt: A *friendly bear **dressed like an explorer** walking in some undiscovered place in the jungle*

Prompt: *A cat dressed like a jedi walk on tatooine holding a **lightsaber**, ultra-realistc hd*

Prompt: *Cute little animals in the forest. In the pond, **a young girl** is touching the gray wolfs chin and they are looking at each other*

Figure 11: Tested the FLUX.1 model with SVDQuant quantization to W4A4 precision(also quantized SVDQuant LoRA branch weight to 6bit). The resolution of the generated image is 256

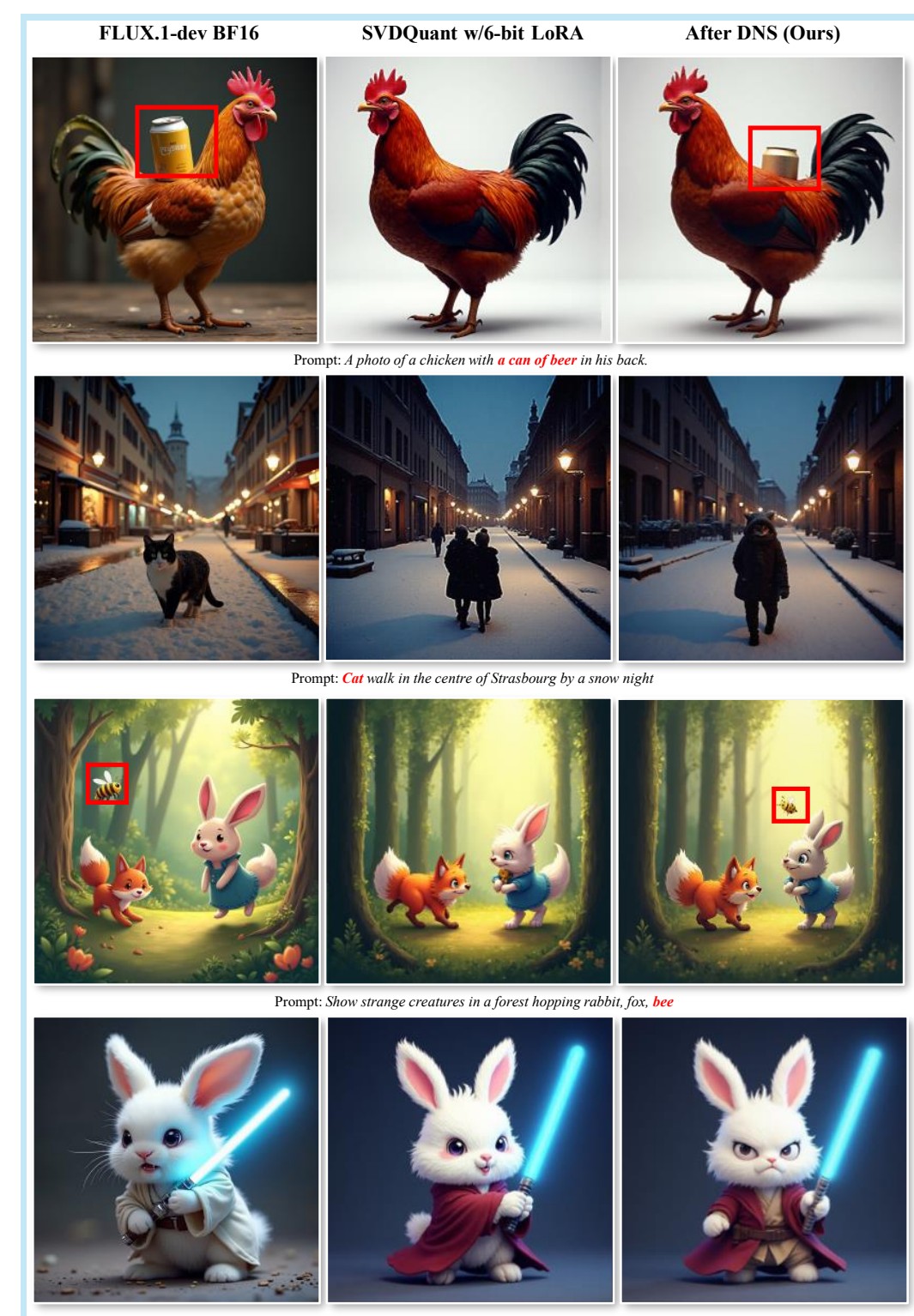

**FLUX.1-dev BF16**      **SVDQuant w/6-bit LoRA**      **After DNS (Ours)**

Prompt: *A photo of a chicken with **a can of beer** in his back.*

Prompt: ***Cat** walk in the centre of Strasbourg by a snow night*

Prompt: *Show strange creatures in a forest hopping rabbit, fox, **bee***

Prompt: *Fluffy white bunny as a Jedi, blue lightsaber, **fierce expression***

Figure 12: Tested the FLUX.1 model with SVDQuant quantization to W4A4 precision(also quantized SVDQuant LoRA branch weight to 6bit). The resolution of the generated image is 1024. It can be seen that we have successfully maintained and recovered some semantics.

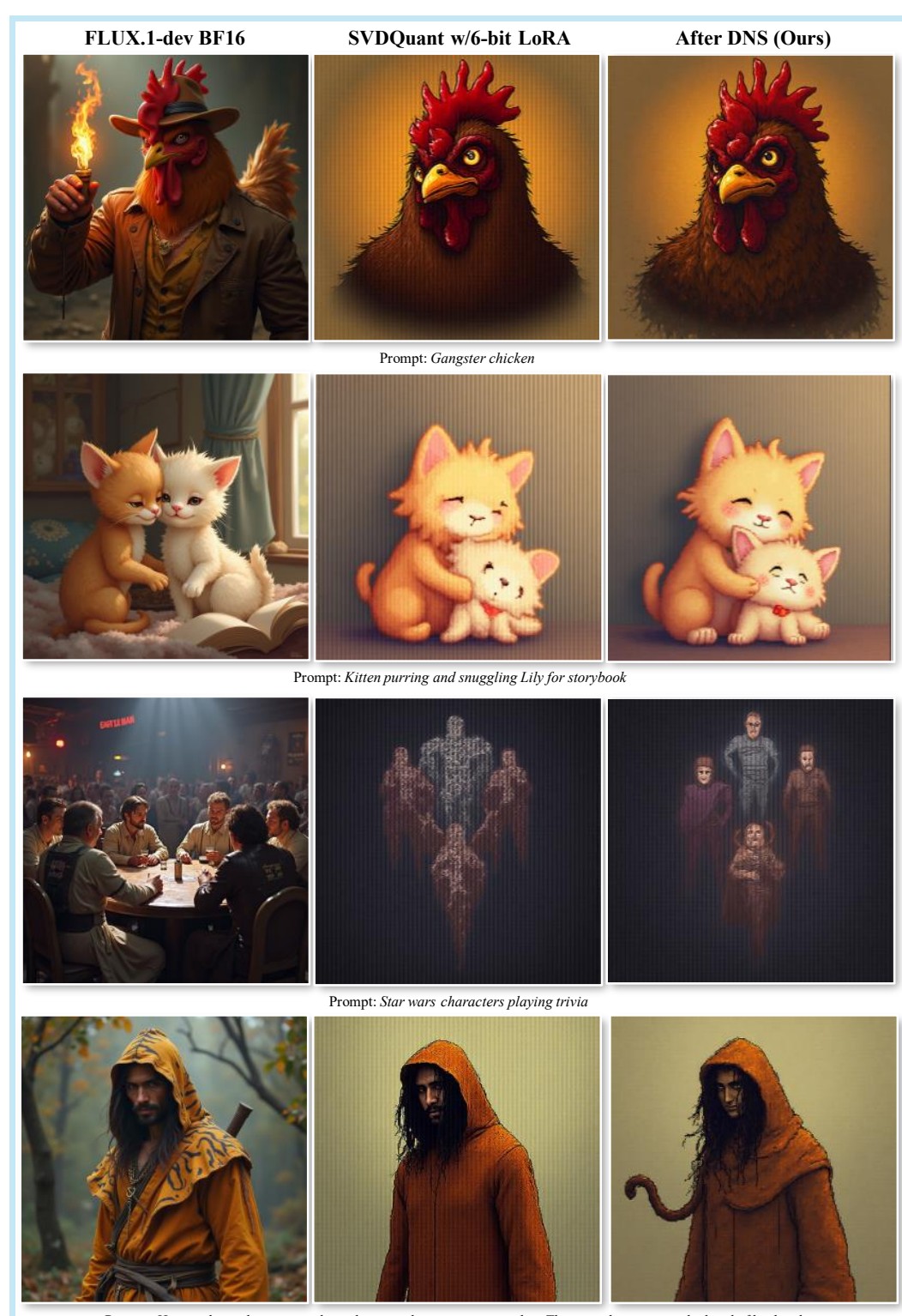

| FLUX.1-dev BF16 | SVDQuant w/6-bit LoRA | After DNS (Ours) |

Prompt: *Gangster chicken*

Prompt: *Kitten purring and snuggling Lily for storybook*

Prompt: *Star wars characters playing trivia*

Prompt: *He was dressed in a tansa dress, his outer hair was a tigers skin, The tiger skin cap was the hood of his head.*
*He had a whip for his upper arm They saw and wanted to see what a stranger saw.*

Figure 13: Tested the FLUX.1 model with SVDQuant quantization to W4A4 precision(also quantized SVDQuant LoRA branch weight to 6bit). The resolution of the generated image is 1024. It can be seen that we have successfully removed some artifacts, but the excessive artifacts introduced by quantization still cannot be completely removed.

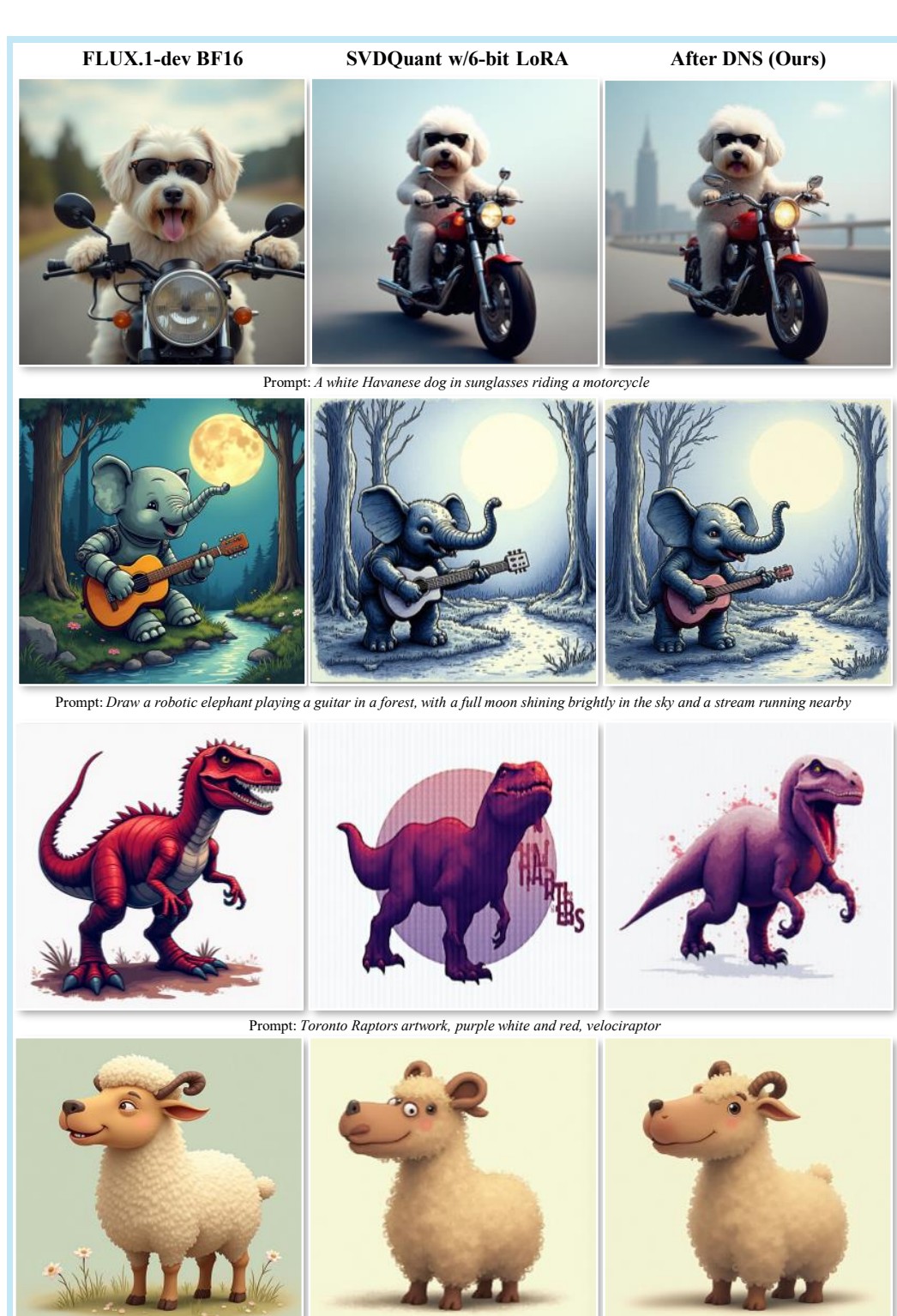

Figure 14: Tested the FLUX.1 model with SVDQuant quantization to W4A4 precision(also quantized SVDQuant LoRA branch weight to 6bit). The resolution of the generated image is 1024. It can be seen that we successfully added some details and improved the image quality.

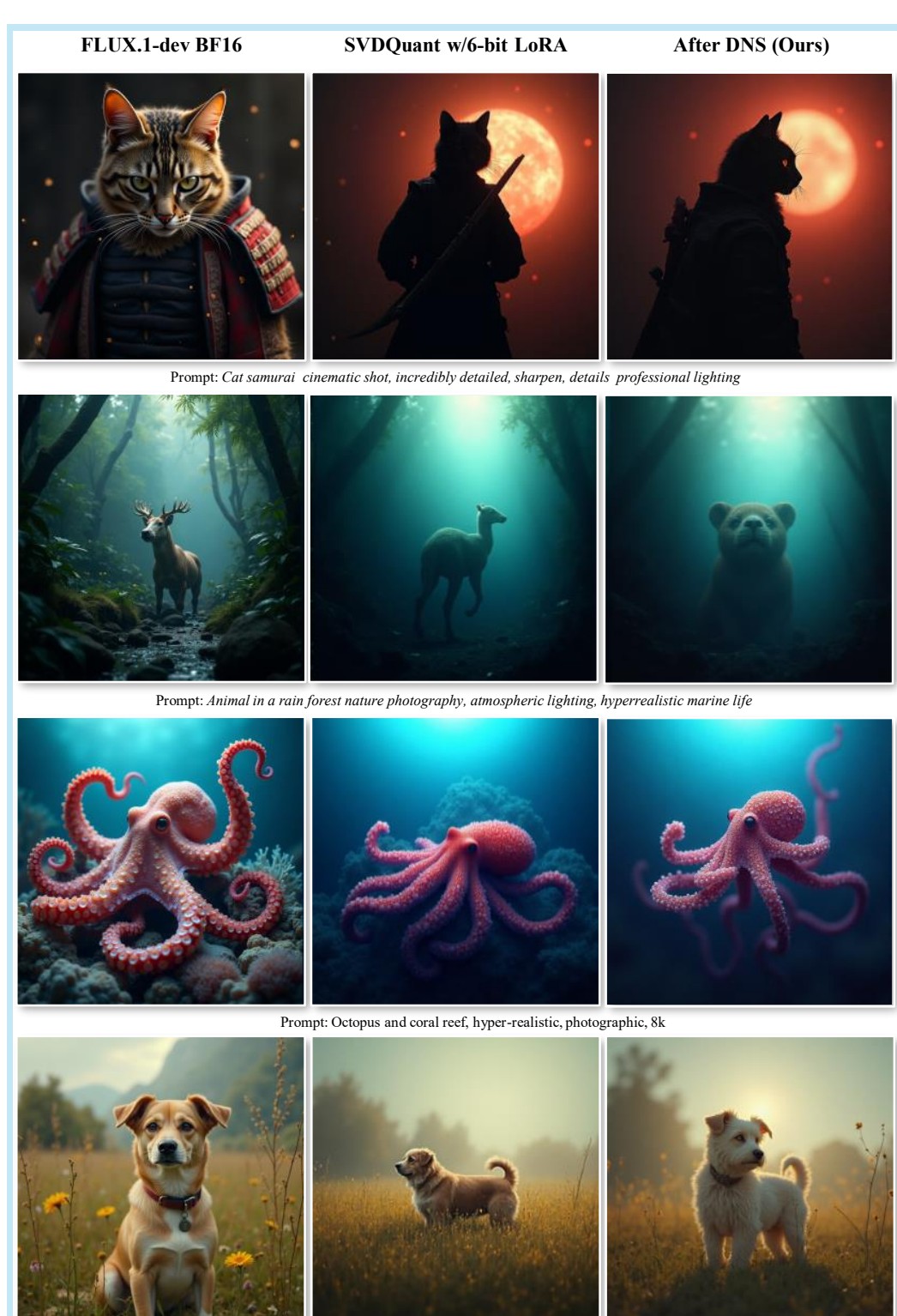

| FLUX.1-dev BF16 | SVDQuant w/6-bit LoRA | After DNS (Ours) |

Prompt: *Cat samurai  cinematic shot, incredibly detailed, sharpen, details  professional lighting*

Prompt: *Animal in a rain forest nature photography, atmospheric lighting, hyperrealistic marine life*

Prompt: Octopus and coral reef, hyper-realistic, photographic, 8k

Prompt: *A A dog on the background of a nature, Wes Anderson style, 4K*

Figure 15: Tested the FLUX.1 model with SVDQuant quantization to W4A4 precision(also quantized SVDQuant LoRA branch weight to 6bit). The resolution of the generated image is 1024. It can be seen that since we are using distribution preservation, some images will also move to other trajectories with the same(as quantized model) quality.

| FLUX.1-dev BF16 | SVDQuant w/ FP LoRA | After DNS (Ours) |
| --- | --- | --- |

Prompt: *Cinematic still. Opossum sitting next to large glass marijuana bong inside messy camper trailer. Setting is nighttime. Ambience is Smokey. Cloud of smoke inside. Hotboxed. 4k. realistic. HDR.*

Prompt: *Cute cow in jacket and hat in winter, reylia slaby style, post processing, ben wooten, soft colors, colorful fauna, cinestill 50d C*

Prompt: *Plecostomus hybrid man picking up PET garbage in Palya del Carmen beach. Sunset Sky*

Prompt: *A woman and a fox dissolve in the rays of the sun, a happy scene full of tenderness, romantic atmosphere, luxurious beauty, exquisite use of additional colors, aesthetic style, unearthly shining light, volumetric lighting*

Figure 16: Tested the FLUX.1 model with SVDQuant quantization to W4A4 precision(maintain SVDQuant LoRA branch precision at W16A16). The resolution of the generated image is 256

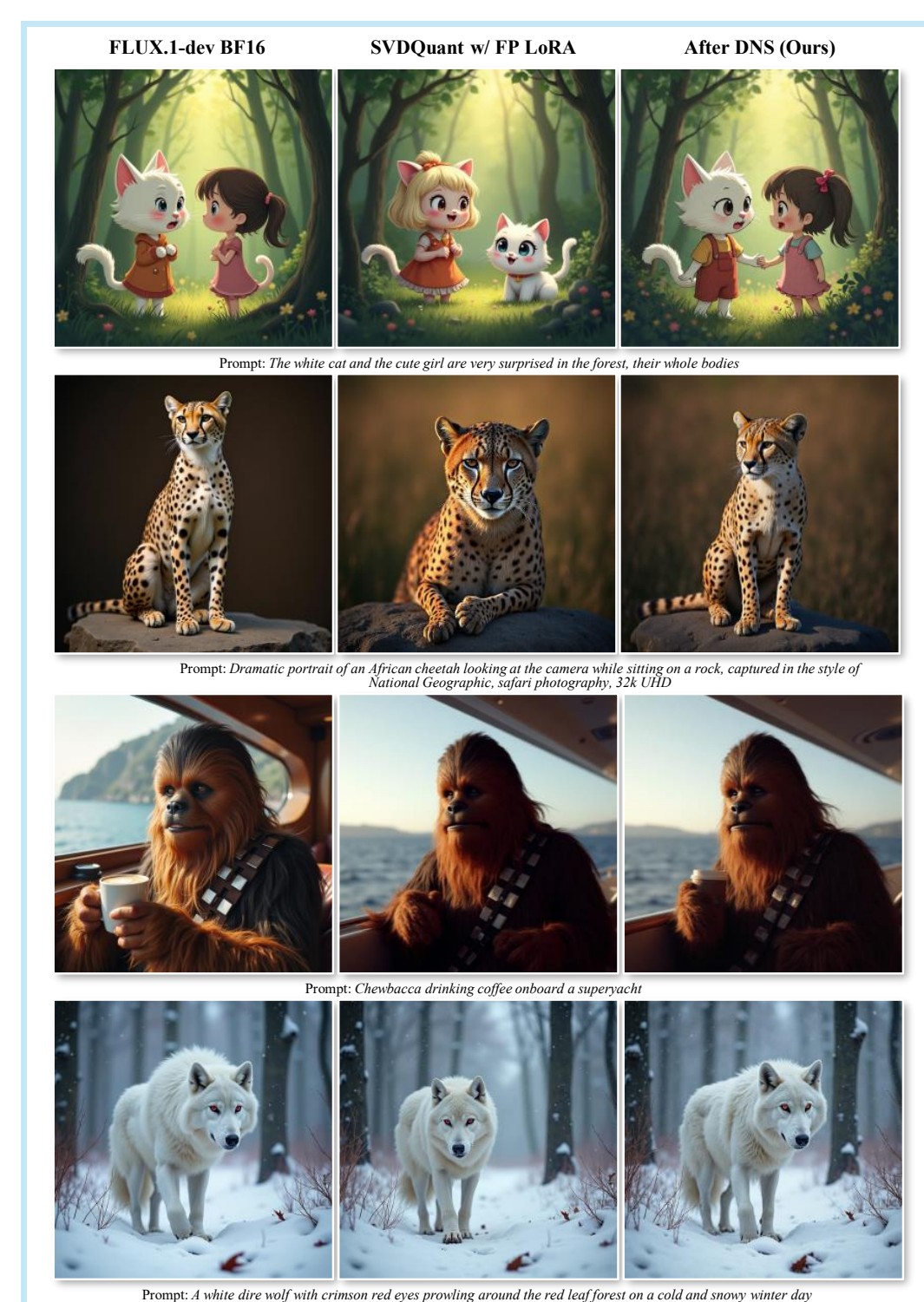

Figure 17: Tested the FLUX.1 model with SVDQuant quantization to W4A4 precision(maintain SVDQuant LoRA branch precision at W16A16). The resolution of the generated image is 1024

Table 9: CIFAR-10 (32×32) generation with TFMQ-DM (W4A8), 20 steps

| Method | FID↓ | sFID↓ | KID×10³ ↓ | IS↑ | PSNR↑ | LPIPS↓ |
|--------|------|-------|-----------|-----|-------|--------|
| FP16 | 7.32 | 7.35 | – | 9.27 | – | – |
| TFMQ-DM | 12.83 | 15.69 | 3.59 | 9.26 | 27.81 | 0.138 |
| *w/ours* | **11.12** | **13.03** | **2.20** | **9.50** | **28.66** | **0.120** |

Table 10: Unconditional ImageNet-64 (64×64) generation with PTQ4DM(W8A8), 20 steps

| Method | FID↓ | sFID↓ | KID×10³ ↓ | IS↑ | PSNR↑ | LPIPS↓ |
|--------|------|-------|-----------|-----|-------|--------|
| FP16 | 17.89 | 10.48 | – | 15.05 | – | – |
| PTQ4DM | 25.36 | 18.13 | 8.12 | 15.10 | 19.88 | 0.232 |
| *w/ours* | **24.32** | **17.65** | **6.35** | 14.69 | 19.71 | 0.241 |

Table 11: CIFAR-10 (32×32) generation with APQ-DM(W6A6), 20 steps

| Method | FID↓ | sFID↓ | KID×10³ ↓ | IS↑ | PSNR↑ | LPIPS↓ |
|--------|------|-------|-----------|-----|-------|--------|
| FP16 | 11.32 | 8.88 | – | 8.50 | – | – |
| APQ-DM | 16.22 | 8.17 | 5.95 | 8.65 | 24.28 | 0.159 |
| *w/ours* | **15.11** | **7.98** | **3.86** | 8.56 | **26.38** | **0.150** |

Table 12: Quantitative metrics with mean ± standard deviation.

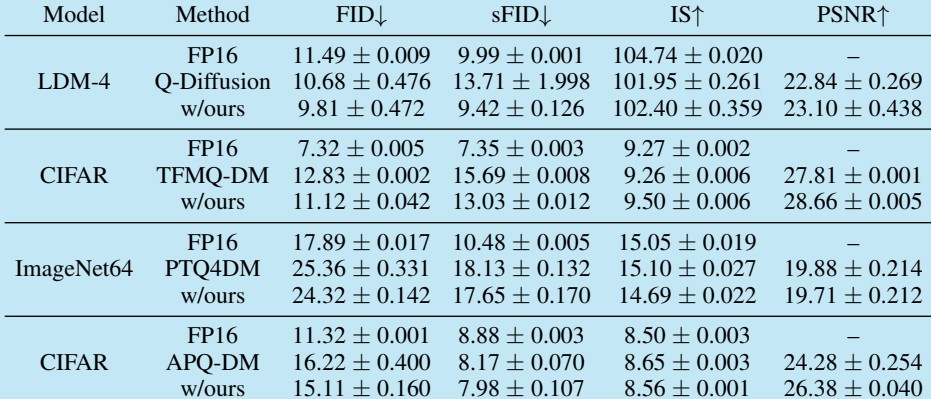

| Model | Method | FID↓ | sFID↓ | IS↑ | PSNR↑ |
|-------|--------|------|-------|-----|-------|
| | FP16 | 11.49 ± 0.009 | 9.99 ± 0.001 | 104.74 ± 0.020 | – |
| LDM-4 | Q-Diffusion | 10.68 ± 0.476 | 13.71 ± 1.998 | 101.95 ± 0.261 | 22.84 ± 0.269 |
| | w/ours | 9.81 ± 0.472 | 9.42 ± 0.126 | 102.40 ± 0.359 | 23.10 ± 0.438 |
| | FP16 | 7.32 ± 0.005 | 7.35 ± 0.003 | 9.27 ± 0.002 | – |
| CIFAR | TFMQ-DM | 12.83 ± 0.002 | 15.69 ± 0.008 | 9.26 ± 0.006 | 27.81 ± 0.001 |
| | w/ours | 11.12 ± 0.042 | 13.03 ± 0.012 | 9.50 ± 0.006 | 28.66 ± 0.005 |
| | FP16 | 17.89 ± 0.017 | 10.48 ± 0.005 | 15.05 ± 0.019 | – |
| ImageNet64 | PTQ4DM | 25.36 ± 0.331 | 18.13 ± 0.132 | 15.10 ± 0.027 | 19.88 ± 0.214 |
| | w/ours | 24.32 ± 0.142 | 17.65 ± 0.170 | 14.69 ± 0.022 | 19.71 ± 0.212 |
| | FP16 | 11.32 ± 0.001 | 8.88 ± 0.003 | 8.50 ± 0.003 | – |
| CIFAR | APQ-DM | 16.22 ± 0.400 | 8.17 ± 0.070 | 8.65 ± 0.003 | 24.28 ± 0.254 |
| | w/ours | 15.11 ± 0.160 | 7.98 ± 0.107 | 8.56 ± 0.001 | 26.38 ± 0.040 |

## A.5 ADDITIONAL EXPERIMENTS RESULTS

**More Backbones Experiments** In addition to the main backbones quantized with Q-Diffusion, PTQ4DiT, and SVDQuant, we further evaluate DNS on other PTQ methods: TFMQ-DM, PTQ4DM, and APQ-DM. We follow exactly the same evaluation protocol and metrics as in Sec. 5.1 and results are shown as Tab. 9-Tab. 11.

**Robustness and Statistical Reliability of Experiments** To further assess the robustness and statistical reliability of our results, we report mean ± standard deviation of three evaluation metrics for representative backbones, as summarized in Tab. 12.

**More Metrics** In addition to FID, sFID, IS, IR, CLIP, and PSNR, we further evaluate Kernel Inception Distance (KID (Bińkowski et al., 2021)) and LPIPS (Zhang et al., 2018). KID is computed

Table 13: Additional experiments with KID and LPIPS across various backbones

| Model | Bit | Method | KID$\times 10^3 \downarrow$ | LPIPS$\downarrow$ |
|---|---|---|---|---|
| LDM-4 | W4A8 | Q-Diffusion | 4.56 | 0.205 |
| | | *w*/PTQD | 3.89 | 0.206 |
| | | *w*/ours | 3.01 | 0.220 |
| DiT-XL | W4A8 | PTQ4DiT | 1.00 | 0.396 |
| | | *w*/PTQD | 2.41 | 0.468 |
| | | *w*/ours | 0.86 | 0.401 |
| FLUX.1 | W4A4 *w*/W16 LoRA | SVDQuant | 0.08 | 0.197 |
| | | *w*/ours | 0.08 | 0.214 |
| | W4A4 *w*/W6 LoRA | SVDQuant | 1.47 | 0.412 |
| | | *w*/ours | 1.22 | 0.431 |
| ImageNet64 | W8A8 | PTQ4DM | 8.12 | 0.232 |
| | | *w*/ours | 6.35 | 0.241 |
| CIFAR-10 | W4A8 | TFMQ | 3.59 | 0.138 |
| | | *w*/ours | 2.20 | 0.120 |
| | W6A6 | APQ | 5.95 | 0.159 |
| | | *w*/ours | 3.86 | 0.150 |

as the squared Maximum Mean Discrepancy between the feature distributions of real and generated images, using the same Inception-V3 backbone as our FID computation. LPIPS measures perceptual distance between image pairs based on deep features from a VGG network, with lower values indicating closer perceptual similarity.

Our primary goal is distribution preservation. Accordingly, FID and KID are especially important, as they quantify distances between feature distributions of generated and FP images under a pre-trained network. By contrast, LPIPS follows the same trend as PSNR: both are pixel- or patch-level similarity metrics that capture per-sample trajectory differences with respect to the FP model and therefore cannot directly validate whether the final outputs achieve better distribution preservation or overall generation quality at the distribution level.

The results, computed under the same settings as in our main experiments for each backbone, are shown in Tab. 13.

To further assess the generality of our approach, we combine it with three representative PTQ methods: TFMQ-DM on CIFAR-10, PTQ4DM on ImageNet-64, and APQ-DM on CIFAR-10. In all three settings, our method consistently improves the generative quality(including FID, KID, etc) over the corresponding PTQ baselines. Concretely, we observe reductions in FID about 4%-13%, in sFID about 2%-17% and in KID about 21%-38% across these experiments. These consistent improvements indicate that our method is effective at preserving distribution under multiple PTQ schemes, leading to quantized model distribution more closely match the FP model distribution.

**Ablation under Other Backbone**   In addition to the LDM-4 analysis already presented, we have extended the ablations to the DiT-XL backbone with PTQ4DiT(Setup is the same as the main result: Using DiT-XL model with PTQ4DiT W4A8 quantization, step=20). We perform the same component-wise ablations as in Sec. 5.3: The results are shown in Tab. 14, Tab. 15, and Fig. 18.

As we can see in Tab. 14, IQR-based variance estimation again performs the best. And in Tab. 15, DNS and especially DCNC consistently improve over the PTQ base model, even for a large-scale backbone like DiT-XL. In Fig. 18, the performance remains stable across a wide range of $W_u$ values, and the same qualitative behavior holds across different architectures and datasets. Specifically, FID

Table 14: Performance comparison of different variance estimation methods in PTQ4DiT.

| Model | Method | FID↓ | IR↑ | PSNR↑ |
|-------|--------|------|------|-------|
| DiT-XL | – | 8.55 | -0.486 | 17.33 |
| | mad | 8.56 | -0.484 | 17.34 |
| | iqr | 8.51 | -0.484 | 17.43 |
| | kus | 8.55 | -0.487 | 17.39 |
| | kde | 8.56 | -0.485 | 17.33 |

Table 15: Effect of different components in PTQ4DiT.

| Model | Method | FID↓ | IR↑ | PSNR↑ |
|-------|--------|------|------|-------|
| DiT-XL | base | 9.83 | -0.508 | 17.42 |
| | +CNC | 14.01 | -0.609 | 16.32 |
| | +DNS | 9.78 | -0.496 | 17.02 |
| | +DCNC | 8.51 | -0.484 | 17.43 |

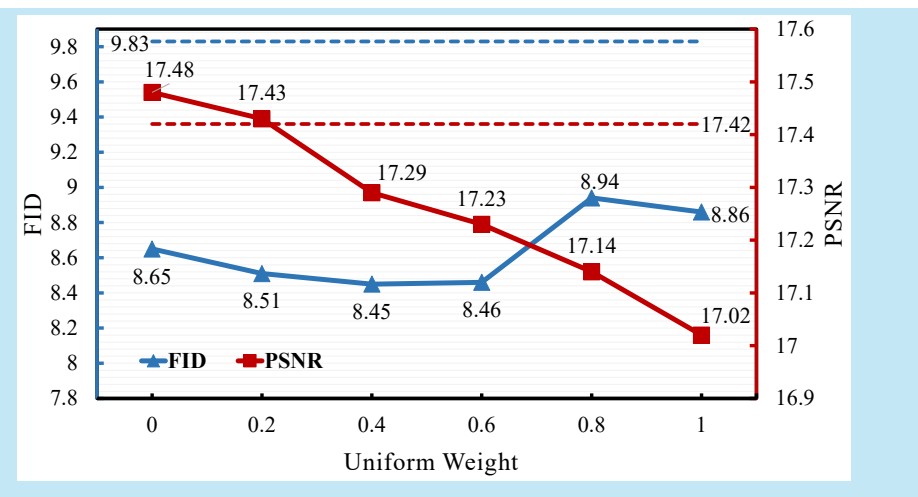

Figure 18: Performance comparison with different strength of correction in PTQ4DiT.

fluctuates somewhat but remains much smaller than before DNS(ours). PSNR decreases slowly as $W_u$ increases. This is consistent with our goal distribution-preserving. Based on these observations, we adopt a single fixed default of $W_u$=0.2 for all models, without any per-model or per-dataset tuning.

**Ablation about More Parameters** We also conducted additional experiments across various sampling steps (10, 20, 40) and guidance scales (0, 3.0, 6.0) and result is shown as Tab. 16.

Our supplementary experiments show that our method consistently maintains superior FID, sFID, IS, and IR scores across all tested configurations. The results demonstrate that our approach remains effective regardless of the number of DDIM steps or guidance scale settings, indicating robust performance across different inference schedules.

The bold part indicates that w/ours is better than other W4A8 indicators.

**Time and Memory Consumption** Regarding our claims of "no extra steps" and "no additional memory", More details are shown in Tab. 17. At inference time, the only additional computation introduced by DNS+DCNC is applying the transformation $\mathcal{T}$ in Eq. 14 and using the pre-computed, timestep-dependent scalar coefficients(Algorithm 2). These operations reduce to elementwise addi-

Table 16: Additional experiments across various sampling steps and guidance scales

| LDM-4 | Method | FID↓ | sFID↓ | IS↑ | CLIP↑ | IR↑ | PSNR↑ |
|---|---|---|---|---|---|---|---|
| scale=3.0 steps=20 | FP16 | 11.49 | 9.99 | 104.74 | 31.91 | 0.134 | — |
| | W4A8 Q-Diffusion | 10.68 | 13.71 | 101.95 | 31.97 | -0.014 | 22.84 |
| | W4A8 w/PTQD | 10.32 | 12.39 | 100.84 | 31.89 | -0.031 | 23.13 |
| | W4A8 w/ours | **9.81** | **9.42** | **102.40** | 31.91 | **0.015** | 23.10 |
| scale=3.0 steps=10 | FP16 | 12.05 | 14.63 | 98.42 | 31.91 | -0.095 | — |
| | W4A8 Q-Diffusion | 11.26 | 21.24 | 92.59 | 31.74 | -0.298 | 24.88 |
| | W4A8 w/PTQD | 13.31 | 38.89 | 85.60 | 31.45 | -0.467 | 20.74 |
| | W4A8 w/ours | **10.43** | **14.94** | **93.29** | **31.76** | **-0.269** | **25.29** |
| scale=3.0 steps=40 | FP16 | 10.66 | 9.15 | 106.32 | 31.81 | 0.215 | — |
| | W4A8 Q-Diffusion | 10.61 | 11.43 | 104.79 | 31.95 | 0.093 | 21.88 |
| | W4A8 w/PTQD | 10.85 | 10.92 | 102.12 | 31.87 | -0.047 | 22.05 |
| | W4A8 w/ours | **9.32** | **8.41** | 104.75 | 31.83 | **0.112** | 21.89 |
| scale=0.0 steps=20 | FP16 | 51.94 | 10.99 | 21.91 | 20.01 | -2.173 | — |
| | W4A8 Q-Diffusion | 59.95 | 20.32 | 20.52 | 20.32 | -2.192 | 26.67 |
| | W4A8 w/PTQD | 58.64 | 20.34 | 20.12 | 20.47 | -2.201 | 25.69 |
| | W4A8 w/ours | **57.47** | **19.39** | **20.66** | 20.30 | **-2.168** | 26.20 |
| scale=6.0 steps=20 | FP16 | 20.89 | 15.31 | 114.55 | 31.98 | 0.382 | — |
| | W4A8 Q-Diffusion | 20.18 | 15.77 | 114.34 | 32.16 | 0.289 | 19.55 |
| | W4A8 w/PTQD | 19.93 | 13.04 | 113.17 | 32.18 | 0.243 | 19.02 |
| | W4A8 w/ours | **18.63** | **10.57** | **115.39** | **32.33** | **0.322** | 19.22 |

Table 17: Overview of calibration and inference overhead for our method (LDM-4 + Q-Diffusion).

| Item | Value |
|---|---|
| Calibration (3,000 images, including FP16 + quantized) | 45 min |
| Calibration statistics only (CPU, excluding sampling) | 214 s |
| Sampling 30k images with quantized model | 5.1 h |
| Extra time for 30k samples with DNS | 5.67 s |
| Peak memory usage | 1.5 GB |
| Additional memory overhead | 320 KB |

tions and multiplications; we do not introduce any extra network forward passes, matrix multiplications, or large activation buffers.

On an A100-40G GPU, for LDM-4 + Q-Diffusion at $256 \times 256$ with 20 DDIM steps, generating 30,000 images, enabling DNS+DCNC incurs only 5.67 seconds of additional wall-clock time compared to PTQ alone. In comparison, generating 30k samples with the quantized model alone already takes about 5.1 hours on our setup, proving that increasing time of our method is almost negligible. The extra runtime buffers required by DNS+DCNC are about 320 KB, and the stored calibration parameters across all timesteps occupy less than 1 KB in total. For calibration, using 3000 images (as in our main experiments), computing the required statistics can be done once on CPU in about 214 seconds. Including both the FP16 and quantized forward passes needed to generate the calibration dataset, the entire calibration procedure takes approximately 45 mins on our setup (A100-40G GPU, LDM-4 + Q-Diffusion at $256 \times 256$ with 20 DDIM steps, sample 3000 images).

