# OpenReview forum: "Absorbing Quantization Error by Deformable Noise Scheduler for Diffusion Models"
_ICLR.cc/2026/Conference — Submitted to ICLR 2026_

### Official Review · Reviewer_uUkK · 2025-10-22

**Soundness:** 4
**Presentation:** 3
**Contribution:** 3
**Rating:** 4
**Confidence:** 3

**Summary:**

This paper proposes a post-training method to absorb quantization error in diffusion models with two main technologies: (1) Distribution-Calibrated Noise Compensation (DCNC). The authors point out that an assumption in previous work -- quantization error is Gaussian -- does not hold. Instead, they use DCNC to statistically correct the mismatch. (2) Deformable Noise Scheduler (DNS). The authors theoretically interpret the quantization noise as a timestep shift in diffusion process and use DNS to schedule the timesteps and compensate for the timestep shift.

**Strengths:**

1. The flow of the paper is smooth, from problem observation to the methods that precisely target the problems.
2. The approaches are inspiring. I especially like the part reinterpreting the quantization noise with diffusion timestep shift.
3. The mitigation of quantization noise kurtosis in Figure 5 is impressive.

**Weaknesses:**

1. The evaluations need more clarification and improvement.
2. Lacking comprehensive discussion compared to other outlier-aware methods.
3. The quality of generated images is not substantially improved.

**Questions:**

1. This paper tackles the quantization error problem -- however, it is anti-intuitive that the results are better than using 16 bits. This important to justify. Is it because quantization error is large even in 16 bits? What if using the plug-and-play approach on FP16 or BF16?
2. Are the evaluation metrics reported averaged over sufficient number of generated images? Error bars should be included for more statistical insights.
3. Mitigating kurtosis of quantization noise seems overlap with outliers handling, as both target distribution tails. Could you discuss and compare the two?
4. Which timestep is Figure 4 at?

---

> ### Author Response · Authors · 2025-11-28
> **Response to Reviewer uUkK(Part 1/3)**
>
> We thank the reviewer for the careful reading, the positive assessment of soundness/presentation, and the constructive suggestions.
>
> - Q1(a): This paper tackles the quantization error problem -- however, it is anti-intuitive that the results are better than using 16 bits.
>
>     A1(a): We understand why this may appear counter-intuitive at first glance. However, this behavior is also observed in prior quantized diffusion work (e.g., **Q-Diffusion**, **PTQD**), where the quantized models occasionally achieve slightly better FID than the FP16 baselines.
>
>     Our analysis is that this phenomenon is **not** due to quantization improving the model capacity, but rather due to the **low-step sampling regime (20 steps)** used in inference-efficient settings. With so few steps, the FP model cannot fully exploit its capacity, and the sampling trajectory becomes dominated by **mismatch between the Gaussian noise assumed by the diffusion process and the actual noise predicted by the network**. In this regime, even FP diffusion models are known to produce suboptimal sample quality.
>
>     Methods such as **Q-Diffusion**, **PTQD**, and our **DNS** act as **noise-distribution corrections**, which reduce this mismatch. As a result, they can occasionally produce **better FID than the FP 20-step baseline**, not because quantization helps, but because the correction improves the sampling distribution.
>
>     **Evidence via kurtosis analysis.**
>
>     To validate this explanation, we measure the kurtosis of the predicted noise under LDM-4 (W8A8 Q-Diffusion) after applying FP, PTQ, CNC (PTQD), and DCNC (DNS). Lower kurtosis indicates a residual distribution closer to the Gaussian assumption.
>
>     | Method | FP  | PTQ | CNC (PTQD) | DCNC (DNS, Ours) |
>     | --- | --- | --- | --- | --- |
>     | Kurtosis  | 0.301 | 0.273 | 0.270 | 0.246 |
>
>     As shown, **both PTQ and PTQD yield lower kurtosis than the original FP model**, and DCNC reduces it further—directly supporting our claim that corrected quantized models can better match the diffusion process’s assumed Gaussian residual than the FP16 model under a 20-step schedule.
>
>     In summary, slightly better performance than FP16 in the low-step regime is an expected and well-understood consequence of **noise-distribution correction**, not an indication that quantization improves inherent model capacity.
>
> - Q1(b): What if using the plug-and-play approach on FP16 or BF16?
>
>     **A1(b):** Our method fundamentally relies on **comparing the predictions of a quantized model against its full-precision counterpart** in order to estimate and correct the residual noise introduced by quantization. When both models are in **FP16 or BF16**, their outputs are already extremely close to FP32, and the discrepancy between the two is nearly negligible. Consequently, the estimated residuals are extremely small, and the resulting DNS/DCNC corrections approach zero.
>
>     In other words, applying the plug-and-play procedure on FP16/BF16 offers **little benefit simply because there is almost no distribution shift to correct**. The method is designed to compensate for quantization-induced distortions, and in high-precision formats such distortions are inherently minimal.

---

> ### Author Response · Authors · 2025-12-03
> **Response to Reviewer uUkK(Part 2/3)**
>
> - Q2: Are the evaluation metrics reported averaged over sufficient number of generated images? Error bars should be included for more statistical insights.
>
>     **A2:** We apologize for not clearly stating the sample sizes in the main text. For completeness, we clarify that all metrics are computed using **30,000 generated images for class-conditional models** and **10,000 generated images for text-to-image models**, which aligns with standard evaluation practice.
>
>     To assess statistical reliability, we now include **error bars** computed over multiple sampling runs. Representative results are shown below:
>
>     |  |  | FID | sFID | IS | PSNR |
>     | --- | --- | --- | --- | --- | --- |
>     | LDM-4 | FP16 | 11.49±0.009 | 9.99±0.001 | 104.74±0.020 | - |
>     |  | Q-Diffusion | 10.68±0.476 | 13.71±1.998 | 101.95±0.261 | 22.84±0.269 |
>     |  | w/ours | 9.81±0.472 | 9.42±0.126 | 102.40±0.359 | 23.10±0.438 |
>     | CIFAR | FP16 | 7.32±0.005 | 7.35±0.003 | 9.27±0.002 | - |
>     |  | TFMQ-DM | 12.83±0.002 | 15.69±0.008 | 9.26±0.006 | 27.81±0.001 |
>     |  | w/ours | 11.12±0.042 | 13.03±0.012 | 9.50±0.006 | 28.66±0.005 |
>     | ImageNet64 | FP16 | 17.89±0.017 | 10.48±0.005 | 15.05±0.019 | - |
>     |  | PTQ4DM | 25.36±0.331 | 18.13±0.132 | 15.10±0.027 | 19.88±0.214 |
>     |  | w/ours | 24.32±0.142 | 17.65±0.170 | 14.69±0.022 | 19.71±0.212 |
>     | CIFAR | FP16 | 11.32±0.001 | 8.88±0.003 | 8.50±0.003 | - |
>     |  | APQ-DM | 16.22±0.400 | 8.17±0.070 | 8.65±0.003 | 24.28±0.254 |
>     |  | w/ours | 15.11±0.160 | 7.98±0.107 | 8.56±0.001 | 26.38±0.040 |
>
>     Across all settings, incorporating our DNS correction not only improves average performance but also **reduces variance**, indicating more stable and reliable sampling. More metrics and experiment details are shown in **Appendix A.5(line 1560, Tab. 12)**
>
>     Further statistical analysis of the residual quantization noise—including error bars for higher-order moments—is provided in **Appendix A.1(line 691,** **Tab. 5)**.
>
> - Q3: Mitigating kurtosis of quantization noise seems overlap with outliers handling, as both target distribution tails.
>
>     **A3:** We appreciate this insightful observation. While both outlier handling and kurtosis reduction involve controlling distribution tails, the mechanisms and objectives are fundamentally different.
>
>     Conventional **outlier-aware PTQ** methods operate on **weights or intermediate activations** by clipping or rescaling extreme values. These approaches suppress large magnitudes at the **parameter or feature level**, and are generally agnostic to the **Gaussian noise model** assumed by the diffusion process.
>
>     In contrast, **DCNC** is designed specifically for diffusion models: it acts **only on the final predicted noise**, and targets the **distribution of the residual quantization error itself**. Rather than truncating tails, DCNC analytically adjusts the residual so that its **higher-order moments (especially kurtosis)** match those of the **ideal Gaussian prior** used in diffusion theory.
>
>     Thus, while both techniques touch distribution tails, **outlier clipping is a parameter-level safeguard**, whereas **DCNC is a principled, distribution-level correction tailored to the diffusion sampling process**.

---

> ### Author Response · Authors · 2025-12-03
> **Response to Reviewer uUkK(Part 3/3)**
>
> - Q4: Which timestep is Figure 4 at?
>
>     A4: Fig. 4 is at timestep t=250. This detail are added in revised version.
>
> - Q5: W1: The evaluations need more clarification and improvement.
>
>     A5: As replied in A2, We will supplement evaluation with details. Meanwhile, we add multiple backbone(TFMQ-DM, PTQ4DM etc.) and additional metrics(LPIPS and KID) in Appendix A.5(**line 1565, Tab. 13**).
>
> - Q6: W2: Lacking comprehensive discussion compared to other outlier-aware methods.
>
>     A6: As replied in A3. We analyzed the comparison between our method and other tail truncation methods.  We also added a conceptual comparison table to show the comparison of our work with previous work , as shown in Appendix A. 2(**line 917, Tab. 7, line 960, Tab. 8**).
>
> - Q7: W3: The quality of generated images is not substantially improved.
>
>     **A7:** We appreciate the reviewer’s comment and would like to clarify the evaluation context. The SVDQuant baseline used in the original Fig. 1 is **unusually strong**(now has updated with a stronger visual impact image, emphasized in red in the revised version) , because it incorporates a **full-precision LoRA branch** into a W4A4 model. This hybrid design brings its performance extremely close to the BF16 model (FID 22.64 → 22.95), leaving **very limited remaining margin for improvement**. Even under this constrained setting, our DNS still improves FID as well as KID and IR.
>
>     Across standard backbones, the gains are much clearer. As shown in **Table 1**, **Fig. 7**, and **Appendix A.2**, our method consistently improves:
>
>     - **distribution-level metrics** (FID, KID, IR—KID added in the revision), and
>     - **visual quality** (Figs. 9–17, with many additional visual samples added).
>
>     These results demonstrate that DNS meaningfully enhances generation quality across a diverse set of models and quantization schemes.
>
>     To further illustrate the impact of the strong baseline, we performed an additional experiment where we **quantized the SVDQuant LoRA branch from FP to W6A16**, restoring a realistic quantization setting with more room for improvement. In this configuration, DNS provides a more substantial uplift:
>
>     |  | FID↓ | KID*1000↓ | CLIP↑ | IR↑ |
>     | --- | --- | --- | --- | --- |
>     | FP16 | 22.64 | - | 29.30 | 0.903 |
>     | SVDQuant w/W6A16 LoRA | 27.16 | 1.47 | 28.89 | 0.693 |
>     | +ours | 26.22 | 1.22 | 29.01 | 0.709 |
>
>     These results show that **when the PTQ baseline leaves a meaningful margin, DNS yields larger and more visible improvements**, while still providing measurable benefits even under extremely strong baselines.
>
>     Additional metrics, analyses, and visual comparisons are provided in:
>
>     - **Appendix A. 4** (line 1134, Figs. 11–15)
>     - **Section 5.2** (line 432, Table 1)

---

### Official Review · Reviewer_kcQd · 2025-10-26

**Soundness:** 3
**Presentation:** 3
**Contribution:** 2
**Rating:** 4
**Confidence:** 4

**Summary:**

The author proposed a plug-and-play approach to improve the quantization
performance of diffusion models through introducing the distribution-Calibrated Noise Compensation (DCNC) that manages to calibrate the distribution to Gaussian, and Deformable Noise Scheduler (DNS) that is aware of the distribution shifts from the quantization. Together, the author claims that the approach preserves the distribution under ubiquitous samplers that fit most situations of multi-step diffusion models. This approach overcomes the limitations of current approaches (reshaping weight distributions, injecting approximated Gaussian noise to absorb residual errors) by ensuring the Gaussian assumption and distribution-awareness. The evaluation results indicate some improvements on strong baselines.

**Strengths:**

1. training-free, plugin-and-play

2. Broadly applicable: works across UNet/DiT/FLUX (incl. deterministic / flow-matching setups) with light calibration and no architecture changes; tangible gains in low-bit regimes (e.g., W4A8).

**Weaknesses:**

1. The improvement seems incremental. The teasing image in Fig. 1 seems to give a rare indication of distributional preservation. In Fig. 2, the FID score of the method is higher than the baseline (22.64 v.s. 22.78).

2. Missing metrics: for distributional matching tasks, LPIPS tells the perceptual distance and is also an important metric; the author should also report this metric in their paper. This metric has been exhaustively reported in SVDQuant, yet is missing in this article. The author should report this to strengthen their contribution.

3. Limited robustness diagnostics: most component/variance-estimator ablations are on a small LDM-4 setup; sensitivity to $W_u$, variance estimator choice, and DNS mapping under different schedulers/backbones (DiT-XL, FLUX) isn't thoroughly characterized.

**Questions:**

1. Is the timestep shift global or per-sample/timestep? Are there any failure cases where DNS degrades diversity or introduces artifacts?

2. For DCNC, any theoretical/experimental justification for the Gaussian claim? Please report the distribution-bias measurement metrics, such as kernel inception distance (KID), across all backbones.

---

> ### Author Response · Authors · 2025-11-28
> **Response to Reviewer kcQd(Part 1/3)**
>
> We thank the reviewer for the careful reading, the positive assessment of soundness/presentation, and the constructive suggestions. We address the questions and weaknesses point by point.
>
> - Q1(a): Is timestep shift global or per-sample/timestep?
>
>     A1(a): Timestep shift is **per-timestep**. In Alg. 1-2, $\alpha, k, \sigma$ are all scalars that depend on t.
>
> - Q1(b): Are there any failure cases where DNS degrades diversity or introduces artifacts?
>
>     **A1(b):** In our experiments, we have not observed any degradation in sample diversity or the introduction of new artifacts. On the contrary, **diversity-sensitive metrics such as FID consistently improve by 8.1–13.4%**, indicating that DNS *enhances* rather than harms generative diversity.
>
>     Qualitatively, DNS does not introduce any characteristic artifacts. In many cases, it **reduces** quantization-induced distortions such as structural inconsistencies and over-smoothed textures (see Fig. 13). Additional visualizations provided in Figs. 1, 11–15 further demonstrate that DNS **preserves semantic content, suppresses artifacts, and often improves detail and variation** in generated samples. These findings are consistent with our theoretical goal: restoring the quantized model’s marginal distribution to match the full-precision diffusion process.
>
>     Additional metrics, analyses, and visual comparisons are provided in:
>
>     - **Appendix A. 4** (line 1134, Figs. 11–15)
>     - **Section 5.2** (line 432, Table 1)
> - Q2(a): For DCNC, any theoretical/experimental justification for Gaussian claim?
>
>     A2(a):  Yes. Our Gaussian assumption for the disentangled residual is supported by both theoretical intuition and empirical evidence.
>
>     **(1) Empirical evidence from the residual distribution.**
>
>     After applying Correlation Noise Correction (Eq. (8)) to isolate the independent residual component, we plot its empirical distribution in **Fig. 4**. Even before correction, this residual already exhibits a **unimodal, bell-shaped profile** closely resembling a Gaussian. This is consistent with a **central-limit-type intuition**: the disentangled residual aggregates many small, weakly correlated perturbations across layers and channels, which naturally tends toward a normal distribution.
>
>     **(2) Quantitative discrepancy motivates DCNC.**
>
>     Although the residual is qualitatively Gaussian-like, quantitative analysis reveals **significant excess kurtosis**, particularly at early and mid timesteps. This mismatch motivates the **derivation of our DCNC kurtosis compensation formula** (Appendix A.1, Eq. 13), which analytically determines the variance of a light-tailed correction term needed to eliminate excess kurtosis and align the residual with a true Gaussian.
>
>     **(3) Higher-order moment analysis confirms improved Gaussianity.**
>
>     In the revised version, we expand the analysis to include **skewness**, **fifth-order central moment**, and **sixth-order excess kurtosis**, across many timesteps. These metrics (Tab. 4, Fig. 8, Appendix A.1 line 691) consistently show that DCNC moves the residual distribution **substantially closer to Gaussian behavior**, not just in the fourth moment but also in higher-order structure.
>
>     Representative timesteps:
>
>     | Timestep | Skewness ($\times10^{-2}$) | $\kappa_x$ | $\kappa_x$ w/ DCNC | $\mu_5$ | $\kappa_6$ | $\kappa_6$ w/ DCNC |
>     | --- | --- | --- | --- | --- | --- | --- |
>     | 950 | 0.26 | -0.07 | -0.07 | $1.03\times10^{-8}$ | -0.47 | -0.47 |
>     | 850 | 0.24 | -0.01 | -0.10 | $0.03\times10^{-8}$ | -0.85 | -0.85 |
>     | 750 | 0.07 | -0.09 | -0.09 | $0.39\times10^{-8}$ | -0.40 | -0.40 |
>     | 650 | -0.51 | 0.21 | -0.02 | $-9.29\times10^{-8}$ | 9.18 | 0.81 |
>     | 550 | -0.94 | 3.30 | 0.08 | $-0.03\times10^{-4}$ | 81.38 | 6.28 |
>     | 450 | -0.88 | 3.34 | 0.14 | $-0.16\times10^{-4}$ | 145.49 | 9.45 |
>     | 350 | -0.62 | 2.33 | 0.15 | $-0.28\times10^{-4}$ | 138.41 | 9.75 |
>     | 250 | -0.39 | 1.78 | 0.14 | $-0.52\times10^{-4}$ | 101.13 | 8.82 |
>     | 150 | -0.36 | 1.33 | 0.10 | $-1.59\times10^{-4}$ | 68.45 | 7.23 |
>     | 50 | -0.51 | 1.11 | 0.08 | $-9.26\times10^{-4}$ | 48.54 | 5.82 |
>
>     Across all timesteps, DCNC reduces higher-order deviations by **1–2 orders of magnitude**, confirming that the corrected residual behaves much more like a Gaussian.
>
>     In sum, the Gaussian claim is supported by (i) empirical bell-shaped distributions, (ii) central-limit intuition for residual formation, and (iii) comprehensive higher-order statistical analysis showing that DCNC brings the residual significantly closer to a Gaussian distribution.

---

> ### Author Response · Authors · 2025-12-03
> **Response to Reviewer kcQd(Part 2/3)**
>
> - Q2(b): Please report distribution-bias measurement metrics, such as kernel inception distance (KID), across all backbones.
>
>     A2(b):  We agree that KID is a valuable metric for assessing distributional alignment. In response to the reviewer’s suggestion, we now report **KID across all evaluated backbones**. In every case, our method **reduces KID relative to the corresponding PTQ baseline**—and often also outperforms PTQD—demonstrating improved preservation of the full-precision model’s generative distribution. The results are summarized below, with additional experiments provided in Appendix A.5(**line 1565, Tab. 13**).
>
>     | Method | KID×\(10^3\)↓ |
>     | --- | --- |
>     | Q-Diffusion | 4.56 |
>     | w/PTQD | 3.89 |
>     | w/ours | 3.01 |
>     | - | - |
>     | PTQ4DiT | 1.00 |
>     | w/PTQD | 2.41 |
>     | w/ours | 0.86 |
>     | - | - |
>     | SVDQuant w/W6 LoRA | 1.47 |
>     | w/ours | 1.22 |
>     | - | - |
>     | PTQ4DM | 8.12 |
>     | w/ours | 6.35 |
>     | - | - |
>     | TFMQ | 3.59 |
>     | w/ours | 2.20 |
>     | - | - |
>     | APQ | 5.95 |
>     | w/ours | 3.86 |
>
>     Across LDM-4, DiT-XL, FLUX.1, UC-UNet variants, and multiple quantization schemes, our method consistently achieves **lower KID**, which directly supports its ability to **preserve the underlying distribution** of the full-precision diffusion model.
>
> - Q3: W1: The improvement seems incremental. Image in Fig. 1 seems to give a rare indication of distributional preservation. In Fig. 2, the FID score of the method is higher than the baseline (22.64 v.s. 22.78).
>
>     **A3:** We appreciate the reviewer’s concern and clarify several important points regarding the perceived incremental improvement.
>
>     **(1) Fig. 1 involved an unusually strong baseline (FLUX.1 + SVDQuant W4A4).**
>
>     The original Fig. 1 (now has updated with a stronger visual impact image, emphasized in red in the revised version) was generated under **FLUX.1 + SVDQuant (W4A4)**. SVDQuant is a particularly strong baseline because it includes a **full-precision LoRA branch**, effectively injecting FP parameters back into a W4A4 model. This dramatically reduces quantization error and leaves **very little remaining margin for additional improvement**.
>
>     Even in this challenging setting, our DNS module still improves multiple metrics—demonstrating that the method remains effective even when the remaining margin is inherently limited.
>
>     **(2) Across standard backbones, improvements are consistent, clear, and multi-metric.**
>
>     In contrast to the W4A4 + FP-LoRA setting, the improvements on conventional diffusion backbones are much larger and highly consistent. As shown in **Table 1**, **Fig. 7**, and **Appendix A.2**, our method produces clear gains across **both metrics (such as FID, KID, IR, Tab. 1) and visual quality** (Figs. 9–17, with additional samples added)
>
>     These results demonstrate that DNS reliably enhances both statistical metrics and perceptual fidelity across diverse architectures.
>
>     **(3) To further address this “remaining margin” limitation, we performed a new experiment quantizing the SVDQuant LoRA branch.**
>
>     To more fairly evaluate DNS, we quantized SVDQuant’s full-precision LoRA branch from FP to **W6A16**, increasing the difficulty of the quantization task and restoring **meaningful room for improvement**. Under this more realistic fully-quantized configuration, DNS again provides **clear improvements**:
>
>     |  | FID↓ | KID*1000↓ | CLIP↑ | IR↑ |
>     | --- | --- | --- | --- | --- |
>     | FP16 | 22.64 | - | 29.30 | 0.903 |
>     | SVDQuant w/W6A16 LoRA | 27.16 | 1.47 | 28.89 | 0.693 |
>     | +ours | **26.22** | **1.22** | **29.01** | **0.709** |
>
>     This experiment forms the basis of the **new Fig. 1**. In the new experiment, it has a stronger visual effect, and we use red to emphasize the improvement, which more accurately illustrates DNS’s distribution-preserving behavior.
>
>     **(4) Extensive results reinforce that the improvements are substantial, not incremental.**
>
>     Additional metrics, analyses, and visual comparisons are provided in:
>
>     - **Appendix 4** (line 1134, Figs. 11–15)
>     - **Section 5.2** (line 432, Table 1)
>
>     Together, these results show that DNS provides **consistent and meaningful improvements**, even under strong baselines where the potential margin for improvement is naturally constrained.

---

> ### Author Response · Authors · 2025-12-03
> **Response to Reviewer kcQd(Part 3/3)**
>
> - Q4: W2: LPIPS is also an important metric; The author should report this to strengthen their contribution.
>
>     A4: We agree that LPIPS is an important perceptual similarity metric, and we now report LPIPS for all evaluated backbones. The results are summarized below, with additional experiments provided in Appendix A.5(**line 1565, Tab. 13**).
>
>     |  Method | LPIPS |
>     | --- | --- |
>     | Q-Diffusion | 0.205 |
>     | w/PTQD | 0.206 |
>     | w/ours | 0.220 |
>     | - | - |
>     | PTQ4DiT | 0.396 |
>     | w/PTQD | 0.468 |
>     | w/ours | 0.401 |
>     | - | - |
>     | SVDQuant | 0.197 |
>     | w/ours | 0.214 |
>     | SVDQuant w/ W6A16 LoRA | 0.412 |
>     | w/ours | 0.431 |
>     | - | - |
>     | PTQ4DM | 0.232 |
>     | w/ours | 0.241 |
>     | - | - |
>     | TFMQ | 0.138 |
>     | w/ours | 0.120 |
>     | - | - |
>     | APQ | 0.159 |
>     | w/ours | 0.150 |
>
>     While we provide LPIPS for completeness, it is worth noting that **LPIPS behaves similarly to PSNR** in that it captures **pixel-level differences** or local perceptual similarity between individual generated samples and references. As such, LPIPS does **not measure distribution alignment or generative diversity**, and is therefore not diagnostic of the core goal of DNS—namely, restoring the **full-precision diffusion distribution** after quantization.
>
>     By contrast, metrics such as **FID**, **KID**, and **Inception Recall (IR)** are explicitly designed to assess **distribution-level fidelity and diversity**, making them far more appropriate for evaluating distribution preservation. On these metrics, our method consistently demonstrates clear improvements across all backbones.
>
> - Q5: W3: Limited robustness diagnostics: most component/variance-estimator ablations are on a small LDM-4 setup; sensitivity to W_u, variance estimator choice, and DNS mapping under different schedulers/backbones (DiT-XL, FLUX) isn't thoroughly characterized.
>
>     A5: We thank the reviewer for the suggestion and have expanded our robustness analysis beyond the small LDM-4 setting. In the revised version, we conduct **the same set of ablations on the much larger DiT-XL backbone** using PTQ4DiT, covering variance estimators, W_u sensitivity, and the effect of DNS/DCNC under different scheduler choices.
>
>     The results are summarized below:
>
>     |  | FID | IR | PSNR |
>     | --- | --- | --- | --- |
>     | - | 8.55 | -0.486 | 17.33 |
>     | MAD | 8.56 | -0.484 | 17.34 |
>     | IQR | 8.51 | -0.484 | 17.43 |
>     | KUS | 8.55 | -0.487 | 17.39 |
>     | KDE | 8.56 | -0.485 | 17.33 |
>     | base |  9.83 | -0.508 | 17.42 |
>     | +CNC | 14.01 | -0.609 | 16.32 |
>     | +DNS | 9.78 | -0.496 | 17.02 |
>     | +DCNC | 8.51 | -0.484 | 17.43 |
>     |Before DNS(ours)| 9.83 | -0.508 | 17.42 |
>     | $W_u$=0 | 8.65 | -0.483 | 17.48 |
>     | $W_u$=0.2 | 8.51 | -0.484 | 17.43 |
>     | $W_u$=0.4 | 8.45 | -0.486 | 17.29 |
>     | $W_u$=0.6 | 8.46 | -0.487 | 17.23 |
>     | $W_u$=0.8 | 8.94 | -0.490 | 17.14 |
>     | $W_u$=1.0 | 8.86 | -0.497 | 17.02 |
>
>     **Key observations (consistent with LDM-4):**
>
>     1. **Variance estimator choice.**
>
>         IQR-based variance estimation again performs the best, mirroring the results from LDM-4. STD and MAD are close but slightly weaker.
>
>     2. **Effectiveness of DNS / DCNC.**
>
>         DNS and especially DCNC consistently improve over the PTQ base model, even for a large-scale backbone like DiT-XL.
>
>     3. **Stability with respect to W_u.**
>
>         We observe the similar insensitivity to W_u as in LDM-4; performance remains stable across a broad range.
>
>         **Thus, we use a single default W_u=0.2 for all backbones and datasets**—no per-model tuning is needed.
>
>
>     These results indicate that our method is **robust across architectures (LDM-4, DiT-XL), variance estimators, and schedulers**, confirming that the core behavior of DNS/DCNC generalizes well beyond the small model used in initial ablations.
>
>     Additional experiments, full tables, and detailed figures are provided in **Appendix 5** (line 1611, Tables 14–15, Fig. 18).

---

### Official Review · Reviewer_MnEb · 2025-10-28

**Soundness:** 3
**Presentation:** 3
**Contribution:** 3
**Rating:** 4
**Confidence:** 3

**Summary:**

This work proposed a distribution-preserving approach that absorbs quantization-induced shifts directly into the sampling process of diffusion/flow models. It proceeds in two stages:
(i) Distribution-Calibrated Noise Compensation, which adds a uniform component to the quantization residual to correct its heavy-tailed deviation;
and (ii) Deformable Noise Scheduler reinterprets quantization as a fractional timestep shift, rewriting the noise schedule so the quantized conditional matches the full-precision marginal.

The method is plug-and-play and works with stochastic and deterministic samplers as well as flow-matching models.

**Strengths:**

1. This work is sufficiently innovative to me. The paper reframes PTQ for diffusion/flow models through a distribution-preserving lens, combining kurtosis-corrected residuals via a calibrated uniform component with treating quantization as a principled timestep shift, extending error absorption beyond stochastic sampling to deterministic samplers and flow-matching.

2. The results of this article are solid, proving the effectiveness of the proposed scheme. Almost all have achieved SOTA in the field of quantization without extra sampling steps, e.g. DiT-XL W4A8 FID 9.83→8.51, even surpassing FP16.

3. The intuition of work seems reasonable to me.

**Weaknesses:**

1. The connection and comparison with previous work is not clear enough.

2. The method of correction does not have a strong theoretical foundation.

3. No quantitative analysis of runtime was performed.

**Questions:**

1. Please position more explicitly what is new beyond: (i) replacing Gaussian residuals with DCNC’s uniform-calibrated residual, and (ii) mapping to a fractional timestep and redesigning the schedule—ideally with a side-by-side derivation comparing your Eqs. (15–20) to PTQD’s update (Eqs. 9–10) and a conceptual table of assumptions/scope (stochastic vs. deterministic/flow).

2. Uniform correction is heuristic; explore richer residual models. DCNC chooses a uniform component to cancel excess kurtosis (Eq. 13/Appendix A.1), which controls a single moment but ignores skew and higher-order structure.

3. The paper states “no extra steps” and “no additional memory,” but there is no quantitative runtime profile for per-step cost or calibration cost. I'm expecting an end-to-end latency and memory for FP16/PTQ/PTQD/Ours on the same GPU, plus the calibration time once vs. reused across runs; include breakdowns to reassure deployability.

---

> ### Author Response · Authors · 2025-11-28
> **Response to Reviewer MnEb(Part 1/4)**
>
> We thank the reviewer for the careful reading, the positive assessment of soundness/presentation, and the constructive suggestions. We address the questions and weaknesses point by point.
>
> - Q1(a). Please position more explicitly what is new beyond: (i) replacing Gaussian residuals with DCNC’s uniform-calibrated residual, and (ii) mapping to a fractional timestep and redesigning the schedule
>
>     A1(a):  Beyond points (i) and (ii), our central contribution is establishing a **new conceptual view of quantization as a timestep shift that preserves the diffusion distribution**.
>
>     More specifically, after applying DCNC, we analytically characterize how quantization perturbs both the **conditional** and **marginal** distributions in the reverse process. We show that the quantized reverse transition remains Gaussian but with an altered variance. This allows us to **map the quantized marginal at timestep t** onto the **full-precision marginal at a shifted timestep $\tau$**, thereby restoring distributional equivalence.
>
>     This perspective fundamentally differs from prior approaches such as PTQD, which only inject residual noise into stochastic samplers without restoring the underlying diffusion distribution.
>
>     Finally, because the mapping relies solely on the Gaussian form of the conditional path, our method is **sampler-agnostic**, requires **no architectural modification**, and integrates with existing PTQ schemes in a completely **plug-and-play** manner.
>
> - Q1(c). A conceptual comparison table
>
>     A1(c): We agree this helps clarify position and scope. In the revision, we have added a concise comparison table:
>
>     | Method | Parameter update?      | Arch. changes or limits?        | Plug-and-play over other PTQ?     | Stoch. | Det. | FM  |
>     |-----------------|-|-|--|--|---|-----|
>     |Q-Diffusion| Yes (Calibration, weight/quant update)           | No             | Not over SVDQuant            | Yes    | Yes  | Yes |
>     | PTQ4DiT      | Yes (Calibration, weight/quant update)           | No; designed for DiT        | Only DiT      | Yes    | Yes  | Yes |
>     | SVDQuant     | Yes (Weight update and FP16 finetune)            | Add FP16 LoRA (Incomplete quant.)        | Not over many model-changing PTQ           | Yes    | Yes  | Yes |
>     | PTQD         | No                            | No                  | Almost Yes                    | Yes    | No   | No  |
>     | **DNS (Ours)** | No                      | No                    | Almost Yes                     | Yes    | Yes  | Yes |

---

> ### Author Response · Authors · 2025-12-03
> **Response to Reviewer MnEb(Part 2/4)**
>
> - Q1(b). A side‑by‑side derivation comparing our Eqs. (15–20) to PTQD’s Eqs. (9–10)
>
>     A1(b):
>
>     The derivations are identical to PTQD up to our Eq. (8), where both methods decompose the quantization error into correlated and uncorrelated components. The key differences arise in how this residual noise is handled.
>
>     **PTQD** proceeds from a **stochastic-sampling perspective**. Starting from the DDPM/DDIM update (our Eq. (5)), PTQD injects the residual noise $\xi_t$ directly into the sampling noise term $\sigma_t^0 z$., yielding the modified stochastic update in their Eqs. (9–10). This approach preserves the stepwise prediction trajectory but does **not** ensure that the resulting marginal distribution remains aligned with the full-precision diffusion process.
>
>     In contrast, **our DNS derivation** adopts a **marginal-distribution perspective**. After DCNC, we examine how quantization perturbs the **entire conditional and marginal distributions** of the reverse process. In our Eq. (17), we show that the quantized reverse step induces a Gaussian with inflated variance. We then construct an explicit **distribution-preserving correspondence** by mapping this quantized marginal onto the full-precision marginal at a **shifted timestep** (our Eq. (20)). This yields a principled timestep-shift interpretation of quantization rather than noise injection.
>
>     A complete side-by-side derivation, aligning our Eqs. (15–20) with PTQD’s Eqs. (9–10), is provided in Appendix A. 2(**line 917, Tab. 7**) for clarity.
>
>     | **PTQD (stochastic-sampling view)** | **DNS (ours, distribution-preserving view)** |
>     | --- | --- |
>     | Both PTQD and our DNS first decompose the quantization error (Eq. 8) as | Both PTQD and our DNS first decompose the quantization error (Eq. 8) as |
>     | $\Delta\epsilon_\theta(x_t,t) = k \epsilon_\theta(x_t,t) + \Delta'\epsilon_\theta(x_t,t).$ | $\Delta\epsilon_\theta(x_t,t) = k \epsilon_\theta(x_t,t) + \Delta'\epsilon_\theta(x_t,t).$ |
>     | This yields the following quantized conditional: | This yields the following quantized conditional: |
>     | $p'\_\theta(x_{t-1}\mid x_t) = \mathcal{N}\big(x_{t-1}; \hat\mu_t(x_t,\hat x_0), \sigma_t^2 I\big) + \xi_t.$ | $p'\_\theta(x_{t-1}\mid x_t) = \mathcal{N}\big(x_{t-1}; \hat\mu_t(x_t,\hat x_0), \sigma_t^2 I\big) + \xi_t.$ |
>     | $\xi_t \sim \mathcal{N}\big(0, C_1^2\sigma_{\epsilon,t}^2 I\big).$ | $\xi_t \sim \mathcal{N}\big(0, C_1^2\sigma_{\epsilon,t}^2 I\big).$ |
>     | where $\xi_t$ is the independent Gaussian component of the quantization noise. | where $\xi_t$ is the independent Gaussian component of the quantization noise. |
>     | Using this decomposition and quantized conditional, PTQD then starts from the standard stochastic sampler (Eq. 5): | Our DNS derivation then starts from the corresponding quantized reverse conditional (Eq. 15), which folds the independent quantization noise into the reverse: |
>     | $x_{t-1} = \frac{1}{\sqrt{\alpha_t}} x_t - \beta_t \frac{1}{\sqrt{1-\bar\alpha_t}} \epsilon_\theta(x_t,t) - \sigma_t^0 z.$ | $p'\_\theta(x\_{t-1}\mid x_t) = q\_\sigma\big(x\_{t-1}\mid x\_t,\hat x'\_0\big) = \mathcal{N}\big( x\_{t-1}; \hat\mu_t(x\_t,\hat x\_0),  (\sigma_t^2 + C_1^2\sigma_{\epsilon,t}^2)I \big).$ |
>     | $z \sim \mathcal{N}(0,I).$ | Thus the induced conditional marginal distribution under quantization is (Eq. 17): |
>     | Substituting the decomposed quantization error into Eq. 5 gives PTQD's stochastic update (Eq. 10): | $q'(x_{t-1}\mid x_0) = \mathcal{N}\big( x_{t-1}; \sqrt{\bar\alpha_{t-1}} x_0,  (1-\bar\alpha_{t-1} + C_1^2\sigma_{\epsilon,t}^2)I \big).$ |
>     | $x_{t-1} = \frac{1}{\sqrt{\alpha_t}} x_t - \beta_t \frac{1}{\sqrt{1-\bar\alpha_t}} \epsilon_\theta(x_t,t) - \sigma_t z - \beta_t c_t \Delta'\epsilon_\theta(x_t,t).$ | DNS then uses Eqs. 18-20 to construct a distributional map that identifies this quantized marginal with an unquantized marginal at a shifted time $\tau$; in particular, Eq. 18 gives |
>     | where $c_t$ is a scalar (a function of $\alpha_t,\bar\alpha_t,k$), and the last two terms $-\sigma_t z - \beta_t c_t \Delta'\epsilon_\theta(x_t,t)$ jointly play the role of the original stochastic noise term $\sigma_t^0 z$ in Eq. 5. Thus $\Delta'\epsilon_\theta$ is treated as an extra stochastic noise term injected into the sampling process. | $q'(x_{t-1}\mid x_0) \longleftrightarrow q\big(C_2 x_\tau\mid x_0\big).$ |
>     |  | and Eq. 20 defines the time shift $\tau$ and the corresponding time-shifted scheduler $\bar\alpha^{(q)}\_{t-1} := \bar\alpha_\tau$, which explicitly constructs a distribution-preserving map that sends the quantized marginal to an original (unquantized) marginal at the shifted time $\tau$. |

---

> ### Author Response · Authors · 2025-12-03
> **Response to Reviewer MnEb(Part 3/4)**
>
> - Q2:  Uniform correction is heuristic; explore richer residual models. DCNC chooses a uniform component to cancel excess kurtosis (Eq. 13/Appendix A.1), which controls a single moment but ignores skew and higher-order structure.
>
>     A2: We appreciate the reviewer’s suggestion and agree that exploring richer models for the residual distribution is a promising direction. Our goal in DCNC, however, is to find a **lightweight, distribution-calibrated correction** that (i) suppresses heavy-tailed behavior introduced by quantization and (ii) preserves efficiency and plug-and-play usability.
>
>     To this end, we conducted additional experiments with several **light-tailed correction distributions** beyond the uniform component used in the main paper:
>
>     ||FID|sFID|IS|
>     |-|-|-|-|
>     |Uniform|9.65|9.33|101.96|
>     |Triangular|9.60|9.14|102.30|
>     |Bimodal Gaussian Mixture|9.77|9.69|101.60|
>     |Generalized Gaussian|9.66|9.35|102.04|
>
>     These results show that DCNC is **not tied to the uniform distribution**—multiple light-tailed families yield comparable improvements. We chose the uniform distribution as the default because it is **extremely efficient to sample** and already provides strong performance(**line 724, Tab. 6**).
>
>     **Higher-order statistical analysis**
>
>     We also performed detailed statistical measurements across timesteps (Tab. 4 and Fig. 8 in the paper). These cover **skewness**, **excess kurtosis**, **fifth-order central moment**, and **sixth-order excess kurtosis**, providing a deeper characterization beyond just controlling the fourth moment.(Based on the definition of kurtosis, we define a sixth-order excess kurtosis $\kappa_6 = \frac{\mu_6}{\sigma^6} - 15$. )
>
>     Representative timesteps are shown below:
>
>     |Timestep|Skewness($\times10^{-2}$)|$\kappa_x$|$\kappa_x$ w/DCNC|$\mu_5$|$\kappa_6$|$\kappa_6$ w/DCNC|
>     |-|-|-|-|-|-|-|
>     |950|0.26|-0.07|-0.07|$1.03\times10^{-8}$|-0.47|-0.47|
>     |850|0.24|-0.01|-0.10|$0.03\times10^{-8}$|-0.85|-0.85|
>     |750|0.07|-0.09|-0.09|$0.39\times10^{-8}$|-0.40|-0.40|
>     |650|-0.51|0.21|-0.02|$-9.29\times10^{-8}$|9.18|0.81|
>     |550|-0.94|3.30|0.08|$-0.03\times10^{-4}$|81.38|6.28|
>     |450|-0.88|3.34|0.14|$-0.16\times10^{-4}$|145.49|9.45|
>     |350|-0.62|2.33|0.15|$-0.28\times10^{-4}$|138.41|9.75|
>     |250|-0.39|1.78|0.14|$-0.52\times10^{-4}$|101.13|8.82|
>     |150|-0.36|1.33|0.10|$-1.59\times10^{-4}$|68.45|7.23|
>     |50|-0.51|1.11|0.08|$-9.26\times10^{-4}$|48.54|5.82|
>
>     DCNC significantly reduces the magnitude of both high-order moments and higher-order kurtosis, making the residual substantially more Gaussian-like. More detailed statistics across all 1,000 timesteps are included in Appendix A. 1(**line 691, Tab. 4, Fig. 8**).
>
> - Q3: There is no quantitative runtime profile for per-step cost or calibration cost. I'm expecting an end-to-end latency and memory for FP16/PTQ/PTQD/Ours on the same GPU, plus the calibration time once vs. reused across runs; include breakdowns to reassure deployability.
>
>     A3: We thank the reviewer for highlighting the importance of quantitative runtime and memory profiling. Our method was explicitly designed to be **deployment-friendly**, introducing only lightweight elementwise operations with **no additional forward passes, no extra matrix multiplications, and no large temporary buffers** (see Algorithm 2). Below we provide detailed latency and memory measurements on the same GPU (A100-40GB).
>
>     **Runtime overhead.**
>
>     When generating **30,000 images** with LDM-4 + Q-Diffusion, DNS adds only **5.67 seconds** of total overhead—corresponding to a **0.03% increase** over a 5.1-hour sampling run. This confirms that the per-step cost of DNS is **negligible** in practice.
>
>     **Memory overhead.**
>
>     The additional runtime buffers required by DNS total only **≈320 KB**, compared to the **1.5 GB peak memory** of the quantized model. We do not modify model weights, activations, or attention maps; only lightweight statistics are stored.
>
>     **Calibration cost (offline).**
>
>     Calibration is performed **once** and reused across all future runs. With 3,000 calibration images, FP16 + quantized forward passes plus CPU-side statistic aggregation take:
>
>     - **45 minutes** end-to-end
>     - **214 seconds** for statistics computation alone (CPU), excluding model inference
>
>     Given that sampling 30k images already takes **5.1 hours**, the one-time calibration cost is small relative to typical deployment workloads.
>
>     |Item|Value|
>     |-|-|
>     |Calibration(3,000 images, including FP16+quantized)|45min|
>     |Calibration statistics only(CPU, except sampling)|214s|
>     |Sampling 30k images with quantized model|5.1h|
>     |Extra time for 30k samples with DNS|5.67s|
>     |Peak memory usage|1.5GB|
>     |Additional memory overhead|320KB|
>
>     The details and experimental settings are summarized in Appendix A. 5(**line 1671, Tab. 17**).

---

> ### Author Response · Authors · 2025-12-03
> **Response to Reviewer MnEb(Part 4/4)**
>
> - Q4: W1: The connection and comparison with previous work is not clear enough.
>
>     A4: As clarified in A1(a)–A1(c), we add a **side‑by‑side derivation** and a **conceptual comparison table** to strengthen the connection and comparison with prior work.
>
> - Q5: W2: The method of correction does not have a strong theoretical foundation.
>
>     A5:  We thank the reviewer for raising this important point. Our correction mechanism is not merely heuristic; it is grounded in a **theoretically justified moment-matching formulation** designed specifically to address the statistical distortions introduced by quantization.
>
>     In **Appendix A.1**, we provide a step-by-step derivation showing that DCNC **provably reduces excess kurtosis** of the quantization residual. Starting from the empirical observation that the independent component of the quantization error exhibits **heavy-tailed, non-Gaussian behavior**, we derive a closed-form solution (Eq. 13) for the amount of light-tailed noise that must be added to make the combined residual match the **kurtosis of a Gaussian (0)**. This provides a precise mathematical criterion—not a heuristic rule—for our correction.
>
>     To further validate this theoretical underpinning, we perform a comprehensive statistical analysis across timesteps:
>
>     - **Skewness (3rd moment)**
>     - **Excess kurtosis (4rd moment)**
>     - **Fifth-order central moment**
>     - **Sixth-order excess kurtosis**
>
>     These measurements (Tab. 4, Fig. 8) show that DCNC not only corrects kurtosis but also significantly reduces other higher-order deviations from Gaussianity. For example, excess kurtosis drops by about **10x-40x**, sixth-order excess kurtosis drops by up to **20×–40×** after correction, and skewness still maintains around zero. This demonstrates that DCNC meaningfully reshapes the residual distribution rather than merely adjusting a single moment.
>
>     In addition, we explore several **alternative light-tailed compensation distributions** (triangular, generalized Gaussian, Gaussian mixture), all of which preserve the same theoretical property of reducing heavy tails. Their comparable performance (Tab. 6) further supports that the mechanism is robust and principled.
>
>     Overall, DCNC provides a **statistically motivated, analytically derived correction** that effectively aligns the quantization residual with the assumptions of the diffusion model’s Gaussian denoising process. It is therefore more than a heuristic—it is a principled approach grounded in distributional analysis and validated through higher-order statistical measurements.
>
> - Q6: W3: No quantitative analysis of runtime was performed.
>
>     A6: As addressed in A3, we now include quantitative end‑to‑end runtime measurements and detailed memory usage statistics, demonstrating that our method is deployable in realistic settings.

---

### Official Review · Reviewer_heeP · 2025-11-01

**Soundness:** 3
**Presentation:** 4
**Contribution:** 2
**Rating:** 2
**Confidence:** 4

**Summary:**

- This paper addresses the challenge of post-training quantization (PTQ) for large diffusion models. Quantization reduces model size and speeds up inference but introduces errors that are particularly damaging to diffusion models, as these errors accumulate over the iterative denoising steps, leading to significant quality degradation.
- The authors identify a key limitation in prior work (like PTQD), which attempts to absorb quantization error as Gaussian noise but is restricted to stochastic samplers and fails for deterministic or modern flow-matching models (e.g., FLUX).
- The paper introduces a novel, plug-and-play framework with two core components:
  - Distribution-Calibrated Noise Compensation (DCNC): This component corrects for the fact that quantization error is not perfectly Gaussian but is often "heavy-tailed." It analytically derives and adds a calibrated amount of uniform noise to the quantization residual, making its statistical properties (specifically, its kurtosis) much closer to a true Gaussian distribution.
  - Deformable Noise Scheduler (DNS): The paper shows that the effect of quantization on the model's output distribution can be interpreted as a timestep shift in the original diffusion process. Instead of adding extra noise, DNS dynamically adjusts the noise schedule (the $\alpha$ values) for the quantized model so that its output distribution at each step aligns with the distribution of the original, full-precision model at a different, "shifted" timestep. This preserves the original generative distribution.
- The method is plug-and-play enhancement approach that works with existing PTQ techniques (e.g., Q-Diffusion, PTQ4DiT, SVDQuant) across various model architectures (LDM, DiT, FLUX.1). It consistently improves quantitative metrics without adding inference latency or memory overhead. Crucially, it works for both stochastic (DDPM) and deterministic (DDIM, Flow Matching) samplers.

**Strengths:**

- The paper is well written.
- The core idea of reinterpreting quantization error as a timestep shift is innovative and provides a principled, unified framework for the problem. It moves beyond heuristic corrections to a distribution-preserving solution.
- A major strength is its compatibility with various sampler types (stochastic, deterministic) and model frameworks (diffusion, flow-matching).
- The method requires no retraining (PTQ), adds no inference overhead, and can be seamlessly integrated with existing quantization methods.
- The paper provides extensive experiments on multiple model backbones (LDM-4, DiT-XL, FLUX.1), bit-precisions (W8A8, W4A8, W4A4), and under different sampling settings (steps, guidance scales).
- The evaluation of the approach shows promising results, outperforming other state-of-the-art works.

**Weaknesses:**

- The method requires a one-time calibration step involving a dataset (3,000 images in the paper) to estimate the noise statistics.
- The method introduces a tunable weight \$W_u\$ for the uniform correction. While the paper selects a value of 0.2, its optimal value might be dataset- or model-dependent, requiring minor hyperparameter tuning for best performance.
- Missing comparison to [1]-[5].
- Minor typo at line 262 ("is" is written twice).

[1] TFMQ-DM: Temporal Feature Maintenance Quantization for Diffusion Models (https://arxiv.org/pdf/2311.16503).
[2] Post-training Quantization on Diffusion Models (https://openaccess.thecvf.com/content/CVPR2023/papers/Shang_Post-Training_Quantization_on_Diffusion_Models_CVPR_2023_paper.pdf).
[3] PQD: Post-training Quantization for Efficient Diffusion Models (https://arxiv.org/abs/2501.00124).
[4] Towards Accurate Post-training Quantization for Diffusion Models (https://openaccess.thecvf.com/content/CVPR2024/papers/Wang_Towards_Accurate_Post-training_Quantization_for_Diffusion_Models_CVPR_2024_paper.pdf).
[5] BiDM: Pushing the Limit of Quantization for Diffusion Models (https://proceedings.neurips.cc/paper_files/paper/2024/file/44b61c5c0ba06d55ab5a1cfb9cfff763-Paper-Conference.pdf).

**Questions:**

How does this method perform when compared to the references mentioned in the weaknesses?

---

> ### Author Response · Authors · 2025-11-28
> **Response to Reviewer heeP (Part 1/2)**
>
> We thank the reviewer for the careful reading, the positive assessment of soundness/presentation, and the constructive suggestions. We address the questions and weaknesses point by point.
>
> - Q1: How does this method perform when compared to the references mentioned in the weaknesses?
>
>     A1: Our method is designed as a **plug-and-play distribution-preserving correction** that can be applied **on top of existing PTQ algorithms** rather than serving as a standalone replacement. Therefore, we evaluate it by integrating our method into several representative PTQ backbones. In the revision, we added full comparisons on top of **TFMQ-DM [1]**, PTQ4DM **[2]**, and APQ-DM **[4]**, reporting the improvement brought by our method in each case.
>
>     Representative quantitative results are shown below.
>
>     |  | FID↓ | sFID↓ | KID*1000↓ | IS↑ | PSNR↑ | LPIPS↓ |
>     | --- | --- | --- | --- | --- | --- | --- |
>     | FP16 | 7.32 | 7.35 | - | 9.27 | - | - |
>     | TFMQ-DM | 12.83 | 15.69 | 3.59 | 9.26 | 27.81 | 0.138 |
>     | w/ours | **11.12** | **13.03** | **2.20** | **9.50** | **28.66** | **0.120** |
>
>     |  | FID↓ | sFID↓ | KID*1000↓ | IS↑ | PSNR↑ | LPIPS↓ |
>     | --- | --- | --- | --- | --- | --- | --- |
>     | FP16 | 17.89 | 10.48 | - | 15.05 | - | - |
>     | PTQ4DM | 25.36 | 18.13 | 8.12 | 15.10 | 19.88 | 0.232 |
>     | w/ours | **24.32** | **17.65** | **6.35** | 14.69 | 19.71 | 0.241 |
>
>     |  | FID↓ | sFID↓ | KID*1000↓ | IS↑ | PSNR↑ | LPIPS↓ |
>     | --- | --- | --- | --- | --- | --- | --- |
>     | FP16 | 11.32 | 8.88 | - | 8.50 | - | - |
>     | APQ-DM | 16.22 | 8.17 | 5.95 | 8.65 | 24.28 | 0.159 |
>     | w/ours | **15.11** | **7.98** | **3.86** | 8.56 | **26.38** | **0.150** |
>
>     To assess generality, we apply our method to **three different PTQ families** (TFMQ-DM, PTQ4DM, APQ-DM) across **two datasets** (CIFAR-10, ImageNet-64). In all cases, our method **consistently improves distribution alignment and generative quality** of the quantized model:
>
>     - **FID** improves by **4%–13%**,
>     - **sFID** improves by **2%–17%**,
>     - **KID** improves by **21%–38%**.
>
>     These gains indicate that our method reliably compensates distributional drift introduced by quantization, helping the quantized model’s generative distribution more closely match the FP16 model’s distribution.
>
>     The details and experimental settings are summarized in Appendix A. 5(**line 1555, Tab. 9-11**).
>
> - Q2 : W1: The method requires a one-time calibration step involving a dataset (3,000 images in the paper) to estimate the noise statistics.
>
>     A2:
>
>     Our method does require a **single offline calibration step** to estimate quantization-induced noise statistics. However, the computational overhead of this step is **small and practical**.
>
>     To quantify this cost, we measured the calibration time on **LDM-4 + Q-Diffusion** using feature maps of size **64×64×3**, **20 timesteps**, and **3,000 calibration images**. On our setup:
>
>     - Computing the required statistics from the saved FP16 and quantized outputs takes **214 seconds on CPU**.
>     - Including both FP16 and quantized forward passes to generate these outputs, the **entire calibration process takes about 45 minutes**.
>
>     Crucially, **almost all** of this time comes from running the FP16 and quantized forward passes—not from our DNS/DCNC computation itself.
>
>     For context, generating **30,000 samples** with the quantized model already requires **≈5.1 hours** on the same hardware. Therefore, the one-time calibration cost is **negligible compared to typical sampling workloads** and is not a bottleneck in practical deployment.
>
>     | Item | Value |
>     | --- | --- |
>     | Calibration (3,000 images, including FP16 + quantized) | 45 min |
>     | Calibration statistics only (CPU, except sampling) | 214 s |
>     | Sampling 30k images with quantized model | 5.1 h |
>     | Extra time for 30k samples with DNS | 5.67 s |
>     | Peak memory usage | 1.5 GB |
>     | Additional memory overhead | 320 KB |
>
>     The details and experimental settings are summarized in Appendix A. 5(**line 1671, Tab. 17**)

---

> ### Author Response · Authors · 2025-12-03
> **Response to Reviewer heeP (Part 2/2)**
>
> - Q3 : W2: The method introduces a tunable weight W_u for the uniform correction. While the paper selects a value of 0.2, its optimal value might be dataset- or model-dependent, requiring minor hyperparameter tuning for best performance.
>
>     A3: We appreciate the reviewer’s concern and agree that any newly introduced hyperparameter warrants careful examination. In practice, however, we find that our method is **not sensitive** to the choice of W_u.
>
>     As shown in **Fig. 6** (LDM-4) and **Fig. 18** in **Appendix A.5** (DiT-XL), the performance remains stable across a wide range of W_u values, and the same qualitative behavior holds across **different architectures and datasets**. Specifically, FID fluctuates somewhat but remains much smaller than before DNS(ours) and PSNR decreases slowly as W_u increases, which is consistent with our goal distribution-preserving. Based on these observations, we adopt a **single fixed default** of W_u=0.2 for **all** models, **without any per-model or per-dataset tuning**.
>     | | | FID | PSNR  |
>     |---|---|---|---|
>     | Q-Diffusion | Before DNS(ours) | 10.68 | 22.84 |
>     |  (LDM-4) | $W_u$ = 0.0      |  9.87 | 23.17 |
>     |  | $W_u$ = 0.2      |  9.81 | 23.10 |
>     |  | $W_u$ = 0.4      |  9.75 | 22.90 |
>     |  | $W_u$ = 0.6      |  9.63 | 22.56 |
>     |  | $W_u$ = 0.8      |  9.47 | 22.05 |
>     | | $W_u$ = 1.0      |  9.42 | 21.17 |
>     | PTD4DiT | Before DNS(ours) |  9.83 | 17.42 |
>     | (DiT-XL)    | $W_u$ = 0.0      |  8.65 | 17.48 |
>     | | $W_u$ = 0.2      |  8.51 | 17.43 |
>     | | $W_u$ = 0.4      |  8.45 | 17.29 |
>     | | $W_u$ = 0.6      |  8.46 | 17.23 |
>     | | $W_u$ = 0.8      |  8.94 | 17.14 |
>     | | $W_u$ = 1.0      |  8.86 | 17.02 |
>
>     All main results in the paper (excluding ablations) use this universal setting of W_u = 0.2.
>
> - Q4 : W3: Missing comparison to [1]-[5].
>
>     As replied in A1, we will add experiments combining our DNS method with these PTQ methods.
>
> - Q5 : W4: Minor typo at line 262 ("is" is written twice).
>
>     Thank you for pointing this out. We will correct this typo in the revised manuscript and carefully proof‑read for similar issues.

---

### Author Response · Authors · 2025-12-03
**Response to Area Chair (Part 1/2)**

Dear AC,

Thank you very much for handling our submission and for constructive reviews.
To better clarify our contributions and changes made in rebuttal, we have prepared a concise summary table.

|Reviewer|Initial rating|Strengths|Questions|Our reply|
|-|-|-|-|-|
|heeP|Overall:2; Soundness:3; Presentation:4; Contribution:2| Well written; Core idea is innovative and provides a principled, unified framework; beyond heuristic corrections to a distribution-preserving solution; Compatibility with various sampler types; No retraining, no inference overhead, seamlessly integrated with existing PTQ|**Q1, W3**: Missing comparison to [1]-[5]; **Q2, W1**: requires a one-time calibration step; **Q3, W2**: introduces a tunable hyperparameter W_u; **Q4, W3**: Minor typo|**A1**: Added full comparisons on top of TFMQ-DM [1], PTQ4DM [2], and APQ-DM [4]; **A2**: Measured time and memory, showing the one‑time calibration cost is negligible for deployment; **A3**: Ablations show a single fixed default W_u works for all models without any per-model or per-dataset tuning; **A4**: Fixed the typo|
|MnEb|Overall:4; Soundness:3; Presentation:3; Contribution:3|Sufficiently innovative; Solid results, proving effectiveness; Without extra sampling steps; Reasonable intuition of work|**Q1, W1**: Comparison with previous work is not clear enough; ideally with a side-by-side derivation and a conceptual table; **Q2**: Uniform correction is heuristic; explore richer residual models; **Q3, W2**: In addition to considering canceling excess kurtosis, skew and higher-order structure should also be taken in DCNC; **Q4, W3:** No quantitative analysis of runtime|**A1**: Added a side‑by‑side derivation and a conceptual comparison table vs prior work; **A2**: Added experiments with several light‑tailed correction distributions; **A3**: Evaluated runtime and memory, confirming no extra forward passes, matrix multiplications, or large temporary buffers|
|kcQd|Overall:4; Soundness:3; Presentation:3; Contribution:2|Training-free, plugin-and-play; Broadly applicable; Light calibration; No architecture changes|**Q1(a)**: Details about timestep shift; **Q1(b)**: Details about diversity and artifacts in DNS; **Q2(a)**: Any theoretical/experimental justification for Gaussian claim for DCNC; **Q3, W1**: Improvement seems incremental; **Q4, Q2(b), W2**: Missing metrics: LPIPS and other distribution-bias metrics such as KID; **Q5, W3**: Limited robustness diagnostics|**A1(a)**: Clarified that the timestep shift is applied at each sampling step; **A1(b)**: Added explanation and visualizations showing DNS increases diversity, suppresses artifacts, preserves semantics, and often improves detail; **A2(a)**: Justified the Gaussian assumption for the disentangled residual with both theoretical arguments and empirical evidence; **A3**: Explained that the seemingly incremental gains only at SOTA are due to limited headroom . However, SOTA retain a FP LoRA with incompletely quantified. Further experiments were conducted by quantizing Lora branch to 6 bits. After use DNS(ours), both metrics and visual quality improve significantly; **A4,A2(b)**: Added KID and LPIPS; **A5:** Added robustness ablations across different backbones.|
|uUkK|Overall:4; Soundness:4; Presentation:3; Contribution:3|Smooth flow of the paper; precisely target the problems; inspiring approaches; especially like the part reinterpreting quantization noise with diffusion timestep shift; Impressive mitigation of quantization noise kurtosis|**Q1:** Please explain why quantization performance is better than full precision; **Q2**: Error bars should be included for more statistical insights; **Q3, Q6, W2**: Please discuss and compare: mitigating kurtosis of quantization noise and outliers handling; **Q5, Q4, W1:** The evaluations need more clarification and improvement; **Q7, W3**: The quality of generated images is not substantially improved.|**A1**: Attribute the improvement to a mismatch between the Gaussian noise assumed by the diffusion process and the actual model‑predicted noise, validated via kurtosis measurements; **A2**: Added metrics with error bars; **A3**: Discussed that outlier clipping is a parameter‑level safeguard, whereas DCNC is a principled distribution‑level correction tailored to diffusion sampling; **A5**: Added missing experimental details and new experiments (more backbones, ablations, metrics, and visualizations); **A7**: Explained that the seemingly incremental gains only at SOTA are due to limited headroom . However, SOTA retain a FP LoRA with incompletely quantified. Further experiments were conducted by quantizing Lora branch to 6 bits. After use DNS(ours), both metrics and visual quality improve significantly.|

---

### Author Response · Authors · 2025-12-03
**Response to Area Chair (Part 2/2)**

Across all reviewers, **Soundness** and **Presentation** are consistently scored **3–4**, indicating consensus that our method is technically solid and clearly presented. The main concern instead lies in the **Contribution** score, where some reviewers characterized our gains as “incremental”. We would like to clarify that this impression arises only in the **SVDQuant** setting: on almost all other backbones, our method brings **substantial improvements**.

The key reason is that the SVDQuant baseline we build on, plugin-and-play, is already very strong, and, importantly, **not fully quantized**: it still keeps a **16‑bit LoRA branch**. While this design preserves performance, it weakens the practical value of PTQ in deployment, since part of the model must still be stored and executed in 16‑bit. To better match realistic deployment and more fairly expose the difficulty of the task, we additionally quantize this LoRA branch to **6‑bit** (**SVDQuant + 6‑bit LoRA**). We find that, even for this SOTA method, quantizing the LoRA branch alone leads to a significant degradation (**FID 22.64 → 27.16**), whereas applying our DNS correction recovers much of the loss and improves **FID to 26.22**, together with consistent gains in **KID, IR**, and other metrics. We have updated **Tab. 1** with these results and added more visual comparisons in **Fig. 1** and **Figs. 11–15**, where improvements from DNS are highlighted in **red**.

We summarize our main revisions below, and briefly state the purpose of each new or updated experiment/clarification:

- **Line 10, Fig. 1; line 416, Fig. 7:** updated visualizations with new qualitative examples to more clearly show how DNS improves visual quality.
- **Sec. 5.2, line 459, Tab. 1:** added a new experiment on **SVDQuant + 6‑bit LoRA** to fully quantize the LoRA branch and demonstrate that DNS can recover a part of the performance loss under this stronger, fully quantized setting.
- **Appendix A.1, line 691, Tab. 4; Fig. 8:** added new experiments analyzing the **skewness, excess kurtosis, and higher‑order central moments of the residuals**, providing empirical support for our statistical modeling and Gaussian residual assumption.
- **Appendix A.1, line 724, Tab. 6:** added experiments comparing **uniform correction** with several other light‑tailed correction distributions.
- **Appendix A.2, line 917, Tab. 7:** added a **side‑by‑side derivation** comparing PTQD and our DNS, to clearly illustrate their theoretical relationship and where our method fundamentally differs.
- **Appendix A.2, line 960, Tab. 8:** added a **concise conceptual comparison table** between DNS and prior PTQ methods, summarizing differences in parameter update method, architecture, Plug-and-play in PTQ, and applicability in different samplers .
- **Appendix A.4, line 1129, Figs. 11–17:** added multiple new **qualitative visualization experiments** under the extended settings, with the improvements from DNS highlighted in red for easier visual inspection.
- **Appendix A.5, line 1555, Tabs. 9–11:** added **more backbone experiments** to show that DNS consistently improves performance across diverse architectures and is not tied to a particular model.
- **Appendix A.5, line 1561, Tab. 12:** added **robustness and statistical reliability experiments** with error bars, reporting variance across runs to strengthen the reliability of our conclusions.
- **Appendix A.5, line 1565, Tab. 13:** added **more evaluation metrics** (KID, LPIPS), providing a more comprehensive assessment beyond previous metrics.
- **Appendix A.5, line 1611, Tabs. 14–15; Fig. 18:** added **ablations under additional backbones** to dissect the contribution of each component of DNS and confirm that our findings generalize.
- **Appendix A.5, line 1671, Tab. 17:** added **time and memory consumption analysis**, quantifying calibration and inference overhead and showing that DNS incurs negligible extra cost.
- **Sec. 1, line 72; Sec. 4.1, line 280; Sec. 5.2, line 459:** improved several **textual and methodological details** by adding missing backbone citations, clarifying the per‑timestep shift implementation, and specifying the number of images used for computing each metric.

---

### Meta-Review · Area_Chair_JZz4 · 2026-01-10

**Summary:**

### Reviewer heeP

1. The method requires a one-time calibration.
    * **Author replies**: The one-time calibration cost is negligible compared to typical sampling workloads and is not a bottleneck in practical deployment.
    * **AC comment**: I think the concern is well addressed. Considering that the calibration is one-time and the cost is
    relatively small (< 1 hour), I think this is not a big issue.
2. The method requires a tunable weight $W_u$, and its optimal value may be dataset or model-dependent.
    * **Author replies**: The authors add experiments with different $W_u$ on two different models and show that
    the method is generally robust to different values of $W_u$. The paper uses the same value for all exps.
    * **AC comment**: I think the concern is well addressed.

### Reviewer MnEb
1. The connection and comparison with previous work is not clear enough
    * **Author replies**: authors added a side-by-side comparison against previous works.
    * **AC comment**: I think the concern is well addressed.
2.  The method of correction does not have a strong theoretical foundation:
    * **Author replies**: The authors make further clarifications on this.
    * **AC comment**: I think the concern is well addressed.
3. No quantitative analysis of runtime was performed
    * **Author replies**: The authors added runtime analysis.
    * **AC comment**: I think the concern is well addressed.

### Reviewer kcQd
1. Improvement is incremental.
    * **Author replies**: The authors argue that SVDQuant is a strong baseline because it includes
    a full-precision branch and is not fully quantized, which reduces quantization errors and leaves
    very little remaining margin for improvement. For other methods, the improvement is significant.
    * **AC comment**:    It's not clear to me how the efficiency of SVDQuant with the full-precision branch compares
    to the cost of the proposed method. From the SVDQuant paper, it seems such a low-rank branch is almost cost-free. So it does not make sense to further quantize it, especially considering that after further quantization, the performance cannot match the original one, even with the proposed method.


2. Missing LPIPS metrics.
    * **Author replies**: The authors added the LPIPS metrics. Similar to PSNR, the proposed methods
    fall behind some baseline methods on LPIPS. The authors argue that LPIPS measures local perceptual
    similarity and does not measure distribution alignment or diversity, and therefore is not diagnostic.
    * **AC comment**: While the argument on LPIPS is reasonable, I think only having distribution level
    metrics may fail to capture the difference in fine-grained details.

3. Limited robustness
    * **Author replies**: The authors added the ablation studies on another backbone DiT-XL and show
    that the conclusions still hold.
    * **AC comment**: I think the concern is addressed.

### Reviewer uUkK

1. The quality of the generated images is not substantially improved
    * **Author replies**: The authors make the same arguments as SVDQuant as mentioned above.
    * **AC comment**: same as replies ot Reviewer kcQd 1.

In general, I think this reviewer's reviews do not provide enough details on his concerns and are very ambiguous.

**Reviewer Concerns:**

Overall, I think most concerns from the reviewers are well addressed. One outstanding concern is:

1. Improvement over SVDQuant is marginal. While the authors argue that SVDQuant is a *unusually strong*
baseline because it adopts a full-precision branch, it's not clear whether such a full-precision branch
will incur a lot of overhead (from the SVDQuant paper, I believe the answer is no). The authors added
a comparison where SVDQuant performs full quantization, which allows the proposed method to bring
more obvious improvement but still fails to match SVDQuant with a full-precision branch. Therefore,
it weakens the claim of the paper, especially considering that the comparison against SVDQuant is
the only one that is done on large-scale text-to-image models.

**Reviewer Scores:**

Reviewer heeP: improve score from 2 to 4 or 6

Reviewer MnEb: improve from 4 to 6

Reviewer kcQd: maintain the score 4

Reviewer uUkK: maintain the score 4


Overall, on the positive side, I think all reviewers appreciate the approach of the method and its novelty, and the rebuttal addressed most of the concerns on presentation, robustness,

On the negative side, they have concerns about the incremental improvement of the paper, especially when compared to SVDQuant on text-to-image models. I believe that the paper needs to do more evaluations on different large-scale text-to-image models (as done in SVDQuant) to demonstrate the effectiveness of the method.

---

### Decision · Program_Chairs · 2026-01-26

Reject